# Spherical arena reveals optokinetic response tuning to stimulus location, size, and frequency across entire visual field of larval zebrafish

Florian A Dehmelt[1†], Rebecca Meier[1†], Julian Hinz[1†‡], Takeshi Yoshimatsu[2], Clara A Simacek[1], Ruoyu Huang[1], Kun Wang[1], Tom Baden[2], Aristides B Arrenberg[1]*

[1]University of Tübingen, Werner Reichardt Centre for Integrative Neuroscience and Institute of Neurobiology, Tübingen, Germany; [2]Sussex Neuroscience, School of Life Sciences, University of Sussex, Sussex, United Kingdom

**\*For correspondence:**
aristides.arrenberg@uni-tuebingen.de

[†]These authors contributed equally to this work

**Present address:** [‡]Friedrich Miescher Institute for Biomedical Research (FMI), Basel, Switzerland

**Competing interests:** The authors declare that no competing interests exist.

## Abstract

Many animals have large visual fields, and sensory circuits may sample those regions of visual space most relevant to behaviours such as gaze stabilisation and hunting. Despite this, relatively small displays are often used in vision neuroscience. To sample stimulus locations across most of the visual field, we built a spherical stimulus arena with 14,848 independently controllable LEDs. We measured the optokinetic response gain of immobilised zebrafish larvae to stimuli of different steradian size and visual field locations. We find that the two eyes are less yoked than previously thought and that spatial frequency tuning is similar across visual field positions. However, zebrafish react most strongly to lateral, nearly equatorial stimuli, consistent with previously reported spatial densities of red, green, and blue photoreceptors. Upside-down experiments suggest further extra-retinal processing. Our results demonstrate that motion vision circuits in zebrafish are anisotropic, and preferentially monitor areas with putative behavioural relevance.

## Introduction

The layout of the retina, and the visual system as a whole, evolved to serve specific behavioural tasks that animals perform to survive in their respective habitats. A well-known example is the position of the eyes in the head which varies between hunting animals (frontal eyes) and animals that frequently need to avoid predation (lateral eyes) (*Cronin et al., 2014*). Hunting animals keep the prey within particular visual field regions to maximise behavioural performance (*Hoy et al., 2016*; *Bianco et al., 2011*; *Yoshimatsu et al., 2020*). To avoid predation, however, it is useful to observe a large proportion of visual space, especially those regions in which predators are most likely to appear (*Smolka et al., 2011*; *Zhang et al., 2012*). The ecological significance of visual stimuli thus depends on their location within the visual field, and it is paralleled by non-uniform processing channels across the retina. This non-uniformity manifests as an *area centralis* or a fovea in many species, which is a region of heightened photoreceptor density in the central retina and serves to increase visual performance in the corresponding visual field regions. Photoreceptor densities put a direct physical limit on performance parameters such as spatial resolution (*Haug et al., 2010*; *Merigan and Katz, 1990*). In addition to these restrictions mediated by the peripheral sensory circuitry, an animal's use of certain visual field regions is also affected by behaviour-specific neural pathways, for example pathways dedicated to feeding and stabilisation behaviour. The retinal and extra-retinal circuit anisotropies can in turn effect a dependence of behavioural performance on visual field location (*Hoy et al.,*

*2016*; *Bianco et al., 2011*; *Murasugi and Howard, 1989*; *Shimizu et al., 2010*; *Yang and Guo, 2013*; *Zimmermann et al., 2018*; *Baden et al., 2013*).

Investigating behavioural performance limits and non-uniformities can offer insights into the processing capabilities and ecological adaptations of animal brains, especially if they can be studied and quantitatively understood at each processing step. The larval zebrafish is a promising vertebrate organism for such an endeavour, since its brain is small and a wide array of experimental techniques are available (*Baier and Scott, 2009*; *McLean and Fetcho, 2011*). Zebrafish are lateral-eyed animals and have a large visual field, which increases during early development and in individual 4-day-old larvae has been reported to cover about 163° per eye (*Easter and Nicola, 1996*), although little is known about interindividual variability. Their retina contains four different cone photoreceptor types (*Nawrocki et al., 1985*), each distributed differently across the retina. UV photoreceptors are densest in the ventro-temporal retina (*area temporalis ventralis*), whereas the red, green, and blue photoreceptors cover more central retinal regions (*Zimmermann et al., 2018*).

Zebrafish larvae perform a wide range of visually mediated behaviours, ranging from prey capture (*Trivedi and Bollmann, 2013*; *Mearns et al., 2020*) and escape behaviour (*Heap et al., 2018*) to stabilisation behaviour (*Kubo et al., 2014*; *Orger et al., 2008*); however, the importance of stimulus location within the visual field for the execution of the respective behaviours has only recently been recognized and is still not well understood (*Hoy et al., 2016*; *Zimmermann et al., 2018*; *Mearns et al., 2020*; *Kist and Portugues, 2019*; *Wang et al., 2020*; *Johnson et al., 2020*; *Lagogiannis et al., 2020*).

During visually mediated stabilisation behaviours, such as optokinetic and optomotor responses, animals move their eyes and bodies, respectively, in order to stabilise the retinal image and/or the body position relative to the visual surround. The optokinetic response (OKR) consists of reflexively executed stereotypical eye movements, in which phases of stimulus 'tracking' (slow phase) are interrupted by quick phases (*Figure 1a*). In the quick phases, eye position is reset by a saccade in the direction opposite to stimulus motion. In humans, optokinetic responses are strongest in the central visual field (*Howard and Ohmi, 1984*). Furthermore, lower visual field locations of the stimulus evoke stronger OKR than upper visual field locations, which likely represents an adaptation to the rich optic flow information available from the structures on the ground in the natural environments of primates (*Murasugi and Howard, 1989*; *Hafed and Chen, 2016*).

In zebrafish larvae, OKR behaviour has been used extensively to assess visual function in genetically altered animals (*Brockerhoff et al., 1995*; *Muto et al., 2005*; *Neuhauss et al., 1999*). OKR tuning to velocity, frequency, and contrast of grating stimuli has been measured (*Clark, 1981*; *Cohen et al., 1977*; *Huang and Neuhauss, 2008*; *Rinner et al., 2005*), and, more recently, zebrafish are used as a model for investigating vertebrate sensorimotor transformations (*Kubo et al., 2014*; *Portugues et al., 2014*). While zebrafish can distinguish rotational from translational optic flow to evoke appropriate optokinetic and optomotor responses (*Kubo et al., 2014*; *Naumann et al., 2016*; *Wang et al., 2019*), it is still unclear which regions of the visual field zebrafish preferentially observe in these behaviours. The aquatic lifestyle, in combination with the preferred swimming depths (*Lindsey et al., 2010*), might cause the lower visual field to contain less relevant information when compared to terrestrial animals. This in turn might result in behavioural biases to other – more informative – visual field regions. A corresponding systematic behavioural quantification in zebrafish or other aquatic species, which would relate OKR behaviour to naturally occurring motion statistics and the underlying neuronal representations in retina and retino-recipient brain structures, has been prevented by technical limitations. Specifically, little is known about (i) the dependence of OKR gain on stimulus location or (ii) on stimulus sizes, (iii) possible interactions between stimulus location, size and frequency, (iv) putative asymmetries between the left and right hemispheres of the visual field, and (v) the relationship between a putative dependence of OKR on stimulus location and zebrafish retinal architecture.

In other species with large visual fields, such as *Drosophila*, full-surround stimulation setups have been designed and used successfully (*Reiser and Dickinson, 2008*; *Kim et al., 2017*; *Maisak et al., 2013*), but to date, none has been used for fish. This is at least partly due to their aquatic environment and the associated difficulties regarding the refraction of stimulus light at the air-water interface. Such distortions of shape can be partially compensated by pre-emptively altering the shape of the stimulus. However, using regular computer screens or video projection, the resulting luminance profiles remain anisotropic unless compensated, potentially biasing the response toward brighter

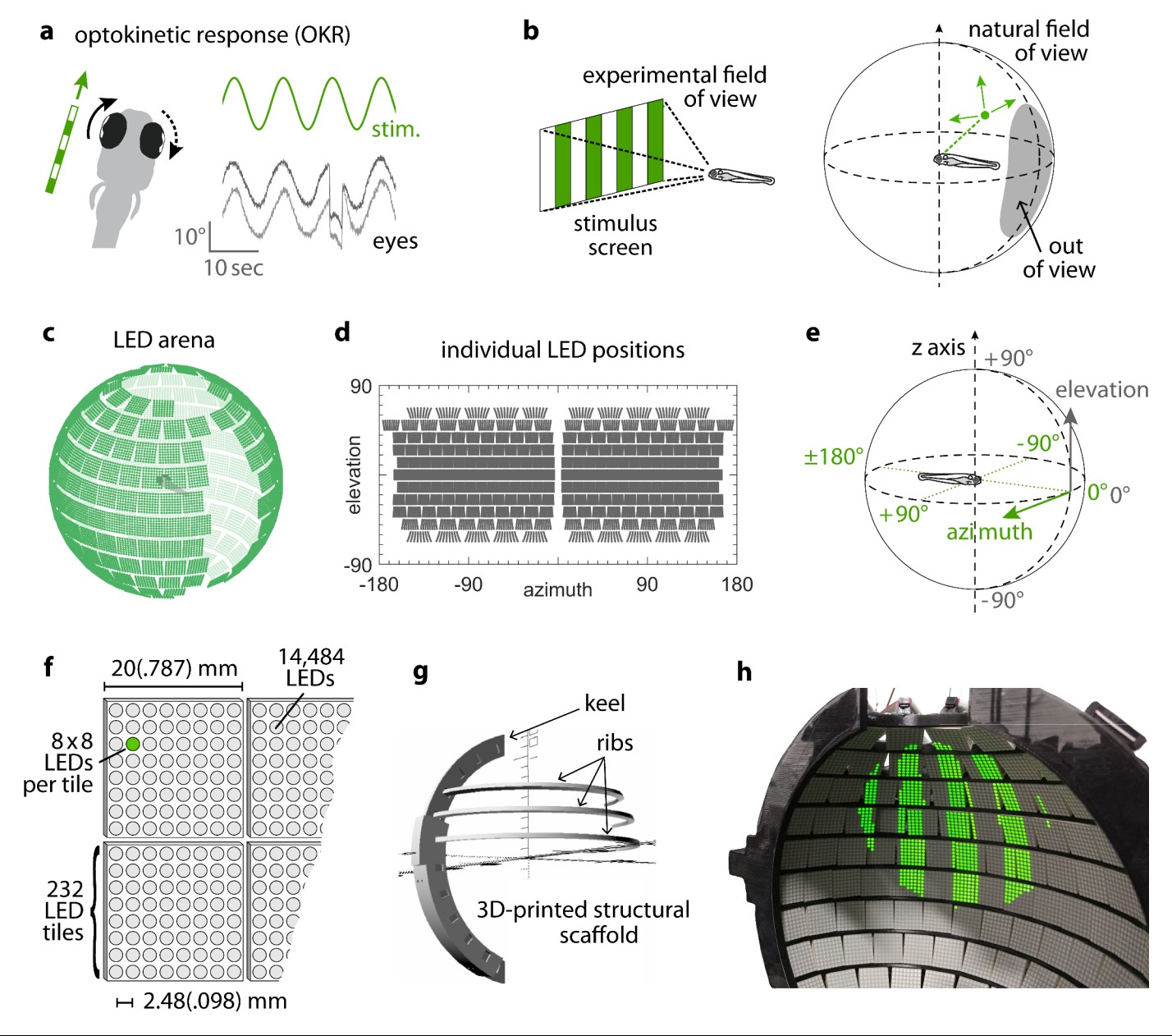

**Figure 1.** Presenting visual stimuli across the visual field. (a) When presented with a horizontal moving stimulus pattern, zebrafish larvae exhibit optokinetic response (OKR) behaviour, where eye movements track stimulus motion to minimise retinal slip. Its slow phase is interrupted by intermittent saccades, and even if only one eye is stimulated (solid arrow), the contralateral eye is indirectly yoked to move along (dashed arrow). (b) Often, experiments on visuomotor behaviour such as OKR sample only a small part of the visual field, whether horizontally or vertically. As different spatial directions may carry different behavioural importance, an ideal stimulation setup should cover all or most of the animal's visual field. For zebrafish larvae, this visual field can be represented by an almost complete unit sphere. (c) We arranged 232 LED tiles with 64 LEDs each across a spherical arena, such that 14,484 LEDs (green dots) covered nearly the entire visual field. (d) The same individual positions, shown in geographic coordinates. Each circle represents a single LED. Each cohesive group of eight-by-eight circles corresponds to the 64 LEDs contained in a single tile. (e) To identify LED and stimulus locations, we use Up-East-North geographic coordinates: Azimuth $\alpha$ describes the horizontal angle, which is zero in front of the animal and, when seen from above, increases for rightward position. Elevation $\beta$ refers to the vertical angle, which is zero throughout the plane containing the animal, and positive above. (f) The spherical arena is covered in flat square tiles carrying 64 green LEDs each. (g) Its structural backbone is made of a 3D-printed keel and ribs. Left and right hemispheres were constructed as separate units. (h) Across 85-90% of the visual field, we can then present horizontally moving bar patterns of different location, frequency and size to evoke OKR.

The online version of this article includes the following source data and figure supplement(s) for figure 1:

**Source data 1.** SCAD files for 3D-printing the arena scaffold.

*Figure 1 continued on next page*

*Figure 1 continued*

**Figure supplement 1.** A spherical LED arena to present visual stimuli across the visual field.
**Figure supplement 2.** Nearest-neighbour distances between LEDs.

locations. Additionally, most stimulus arenas cannot easily be combined with the recording of neural activity, for example, via calcium imaging, as stimulus light and calcium fluorescence overlap in both the spectral and time domains. These challenges must be overcome to enable full-field visual stimulation in zebrafish neurophysiology experiments (*Figure 1b*). At least one existing solution for stimulating and tracking freely moving zebrafish supports unusually large stimuli, although despite its versatility, it still only covers part of the visual field and does not address the remaining issues of total internal reflection and the interoperability with laser-scanning microscopes (*Stowers et al., 2017*).

Here, we present a novel visual stimulus arena for aquatic animals, which covers almost the entire surround of the animal, and use it to characterise the anisotropy of the zebrafish OKR across different visual field locations as well as the tuning to stimulus size, spatial frequency and leftside versus rightside stimulus locations. We find that the OKR is mostly symmetric across both eyes and driven most strongly by lateral stimulus locations. These stimulus locations approximately correspond to a retinal region of increased photoreceptor density. By rotating the experimental setup and/or the animal, our control experiments revealed that additional extra-retinal determinants of OKR drive exist as well. Our characterisation of OKR drive across the visual field will help inform bottom-up models of the vertebrate neural pathways underlying the optokinetic response and other visual behaviour.

## Results

### Spherical LED arena allows presentation of stimuli across the visual field

By combining 3D printing with electronic solutions developed in *Drosophila* vision research, we constructed a spherical stimulus arena containing 14,848 individual LEDs covering over 90% of the visual field of zebrafish larvae (*Figure 1c*, Materials and methods section on coverage). Using infrared illumination via an optical pathway coupled into the sphere (*Figure 1—figure supplement 1a–b*), we tracked eye movements of larval zebrafish during presentation of visual stimuli (*Dehmelt et al., 2018*).

To avoid stimulus aberrations at the air-to-water interface, we designed a nearly spherical glass bulb containing fish and medium (*Figure 1—figure supplement 1c–d*). With this design, stimulus light from the surrounding arena is virtually not refracted (light is orthogonal to the air-to-water interface) and reaches the eyes of the zebrafish larva in a straight line. Thus, no geometric corrections are required during stimulus design (*Source code 1*), and stimulus luminance is expected to be nearly isotropic across the visual field. We additionally designed the setup to minimise visual obstruction and developed a new embedding technique to immobilise the larva at the tip of a narrow glass triangle (see Materials and methods). In almost all possible positions, fish can thus perceive stimuli without interference (*Wang et al., 2021*). The distance between most of the adjacent LED pairs is smaller than the photoreceptor spacing in the larval retina (*Haug et al., 2010*; *Tappeiner et al., 2012*), resulting in a good spatial resolution across the majority of the spherical arena surface (*Figure 1—figure supplement 2*, see Materials and methods section on resolution). As flat square LED tiles cannot be perfectly arranged on a spherical surface (*Supplementary file 1A*), small triangular gaps are unavoidable. More importantly, several gaps in LED coverage, resulting from structural elements of the arena, were restricted mainly to the back, the top, and bottom of the animal. The 'keel' behind and in front of the fish supports the horizontal 'ribs', and the circular openings in the top and bottom accommodate the optical path for eye tracking or scanning microscopy (also see discussion of arena geometry in our data repository [*Dehmelt et al., 2020*; https://gin.g-node.org/Arrenberg_Lab/spherical_arena]).

## Stimulus position dependence of the optokinetic response

Horizontally moving vertical bars reliably elicit OKR in zebrafish larvae (*Beck et al., 2004*). We used a stimulus which rotated clock- and counter-clockwise with a sinusoidal velocity pattern (velocity amplitude 12.5 degree/s, frequency of the velocity envelope 0.1 Hz, spatial frequency 0.06 cycles/degree, *Figure 2a*). OKR performance was calculated by measuring the amplitude of the resulting OKR slow-phase eye movements after the saccades had been removed (*Figure 2b*, *Figure 2—figure supplement 1*, *Source code 1C*, Materials and methods). The OKR gain then corresponds to the speed of the slow-phase eye movements divided by the speed of the stimulus (which is equivalent to the ratio of the eye position and stimulus position amplitudes). In addition to traditional full-field stimulation, our arena can display much smaller stimuli in different parts of the visual field. These novel stimuli evoked reliable OKR even at remote stimulus locations (*Figure 2c*, *Figure 2—figure supplement 2*), and thus allowed us to investigate previously inaccessible behavioural parameters. We excluded any trials from data analysis that showed other behaviours (e.g. drifts and ongoing spontaneous eye movements) superimposed on OKR (*Figure 2—figure supplement 2*). In addition to the characteristic eye traces (*Figure 2c*), Bode plots of the magnitude and phase shift relative to the stimulus qualitatively match previously reported zebrafish response to full-field stimulation (*Figure 5—figure supplement 1*, *Beck et al., 2004*), further confirming that the behaviour we observe is indeed OKR.

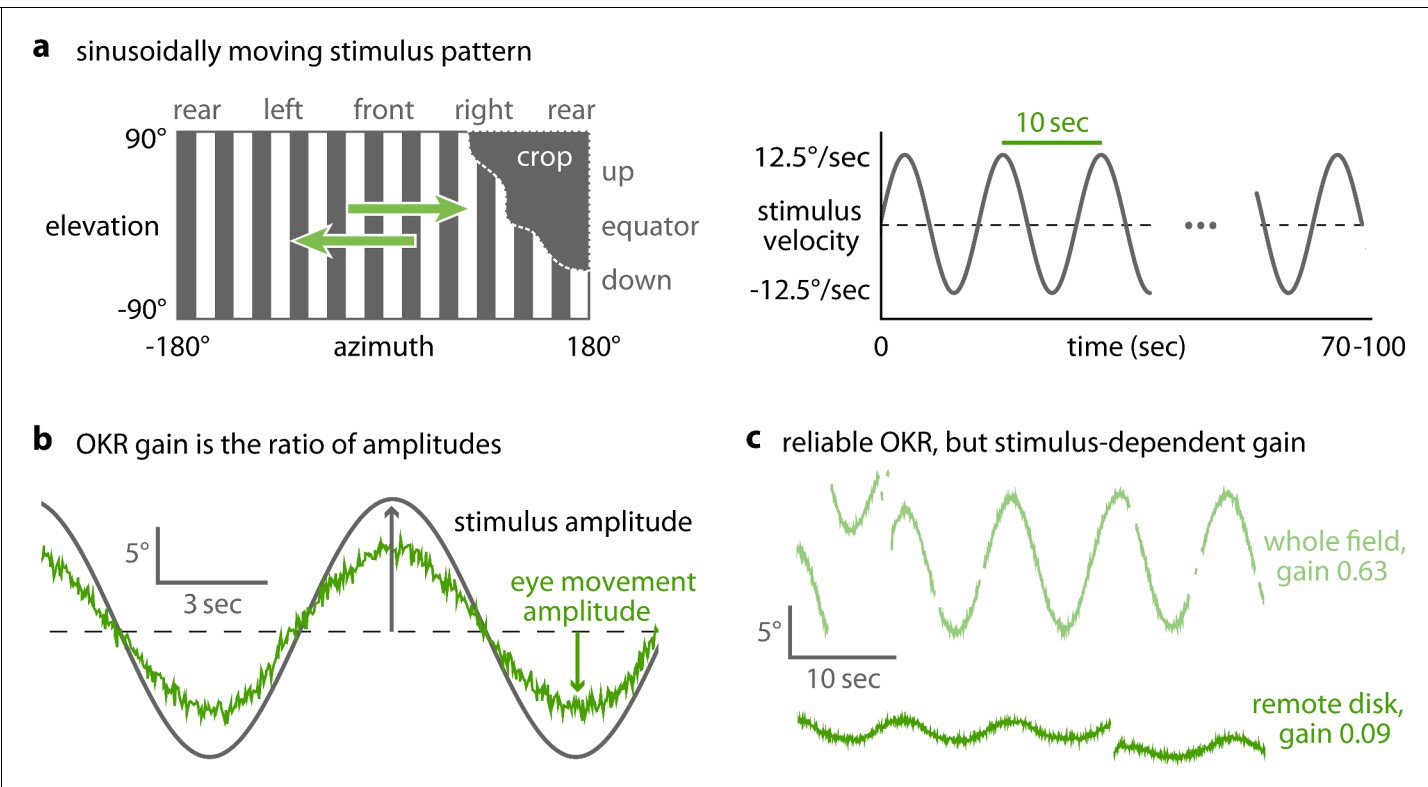

**Figure 2.** OKR gain is inferred from a piece-wise fit to the slow phase of tracked eye movements. (**a**) We present a single pattern of horizontally moving bars to evoke OKR and crop it by superimposing a permanently dark area of arbitrary size and shape (left). Its velocity follows a sinusoidal time course, repeating every 10 s for a total of 100 s for each stimulus phase (right). (**b**) OKR gain is the amplitude of eye movement (green trace) relative to the amplitude of the sinusoidal stimulus (grey trace). The OKR gain is often well below 1, e.g. for high stimulus velocities as used here (up to 12.5˚/s). (**c**) Even small stimuli are sufficient to elicit reliable OKR, although gains are low if stimuli appear in a disfavoured part of the visual field. Shown here are responses to a whole-field stimulus (top) and to a disk-shaped stimulus in the extreme upper rear (bottom, *Figure 2—figure supplement 2*).

The online version of this article includes the following figure supplement(s) for figure 2:

**Figure supplement 1.** Saccades are cropped before OKR gain is measured.

**Figure supplement 2.** Even at remote stimulus locations, fish exhibit reliable OKR behaviour; trials with mixed behaviour are excluded.

To quantify position tuning, we cropped the presented gratings (*Figure 2a*) to a disk-shaped area of constant size, centred on one of 38 nearly equidistant parts of the visual field (*Figure 3a*, *Supplementary File 1B*, *Video 1*, *Figure 3—video 1*). The distribution of positions was symmetric between the left and right, upper and lower, as well as front and rear hemispheres, with some stimuli falling right on the edge between two hemispheres. As permanent asymmetries in a stimulus arena or in its surroundings could affect OKR gain, we repeated our experiments in a second group of larvae after rotating the arena by 180 degrees (*Figure 4a–b*), then matched the resulting pairs of OKR gains during data analysis (see Materials and methods, *Figure 4e–f*). Any remaining asymmetries in the OKR distributions should result from biological lateralisation, if indeed there exists such consistent lateralisation across individuals.

To overcome our spatially discrete sampling, we then fit our data with a bimodal function comprised of two Gaussian-like two-dimensional distributions on the stimulus sphere surface (see Materials and methods, *Source code 1D*), to determine the location of highest OKR gain evoked by ipsilateral stimuli and contralateral stimuli, respectively. We observed significantly higher OKR gains in response to nearly lateral stimuli, and lower gains across the rest of the visual field (*Figure 3b–e*). OKR was strongest for stimuli near an azimuth of −82.5 degrees and an elevation of 5.1 degrees for the left side (in body-centred coordinates), as well as 81.7 and 1.6 degrees for the right side – slightly rostral of the lateral meridian, and slightly above the equator. In the nasal visual field (binocular overlap) the OKR gain was relatively high, but still lower than for lateral visual field locations. Note that due to the fast stimulus speeds, the absolute slow phase eye velocities were high, while the OKR gain was relatively low. We chose such high stimulus speeds to minimise the experimental recording time needed to obtain reliable OKR measurements for each visual field location.

As our stimulus arena is not completely covered by LEDs (*Figure 1c*, *Figure 1d*), some areas remain permanently dark. These could interfere with the perception of stimuli presented on adjacent LEDs. This is especially relevant as LED coverage is almost perfect for some stimulus positions (near the equator), whereas the size of triangular holes increases at others (towards the poles). We thus performed control experiments comparing the OKR gain evoked by a stimulus in a densely covered part of the arena to the OKR gain evoked by same stimulus, but in the presence of additional dark triangular patches (*Figure 3—figure supplement 1a*). We found no significant difference in OKR gain (*Figure 3—figure supplement 1c*, t-test, p>0.05). Additionally, we performed another series of control experiments using a dark shape mimicking the dark structural elements, the front 'keel' of the arena (*Figure 3—figure supplement 1b*). Again, we found no difference in OKR gain (*Figure 3—figure supplement 1c*, t-test, p>0.05), and thus ruled out that position dependence data was corrupted by incomplete LED coverage. Since the eyes were moving freely in our experiments, the range of eye positions during OKR, or so-called beating field (*Schaerer and Kirschfeld, 2000*), could have changed with stimulus position. We found that animals instead maintained similar median horizontal eye positions (e.g. left eye: −83.7 ± 1.8 degrees, right eye: 80.3 ± 1.9 degrees, average median ± standard deviation of medians, n = 7 fish, *Figure 3—figure supplement 2*) even for the most peripheral stimulus positions.

A priori, it is unclear whether the sampling preference originates from the peculiarities of the sensory periphery in the eye, or the behavioural relevance inferred by central brain processing. The former would establish stimulus preference based on its position relative to the eye and, by extension, its representation on specific parts of the retina (i.e. an eye-centered reference frame). The latter would establish stimulus preference in body- or head-centered reference frames or based on stimulus positions relative to environment (a world-centered reference frame). A world-centered reference frame is useful, if stimuli in different environmental locations have different behavioural relevance (such as a predator approaching from the water surface). To start distinguishing these possible effects in the context of OKR, as well as to reveal any stimulus asymmetries accidentally introduced during the experiment, we performed control experiments with larvae embedded upside-down (i.e. with their dorsum towards the lower pole of the arena, *Figure 4c*). Note that such an upside-down state can occur when the fish loses balance in natural behaviour and would normally be counteracted by the animals' self-righting reflex. As a result of upside-down embedding, world-centred and fish-centred coordinate systems were no longer identical, in that 'up' in one is 'down' in the other, and 'left' becomes 'right'. See *Supplementary file 1C* for a summary of these changes across experiments, and Materials and methods for notes on head- and retina-centred coordinate systems. To facilitate comparisons across embedding types, all positions from here onwards are given relative to

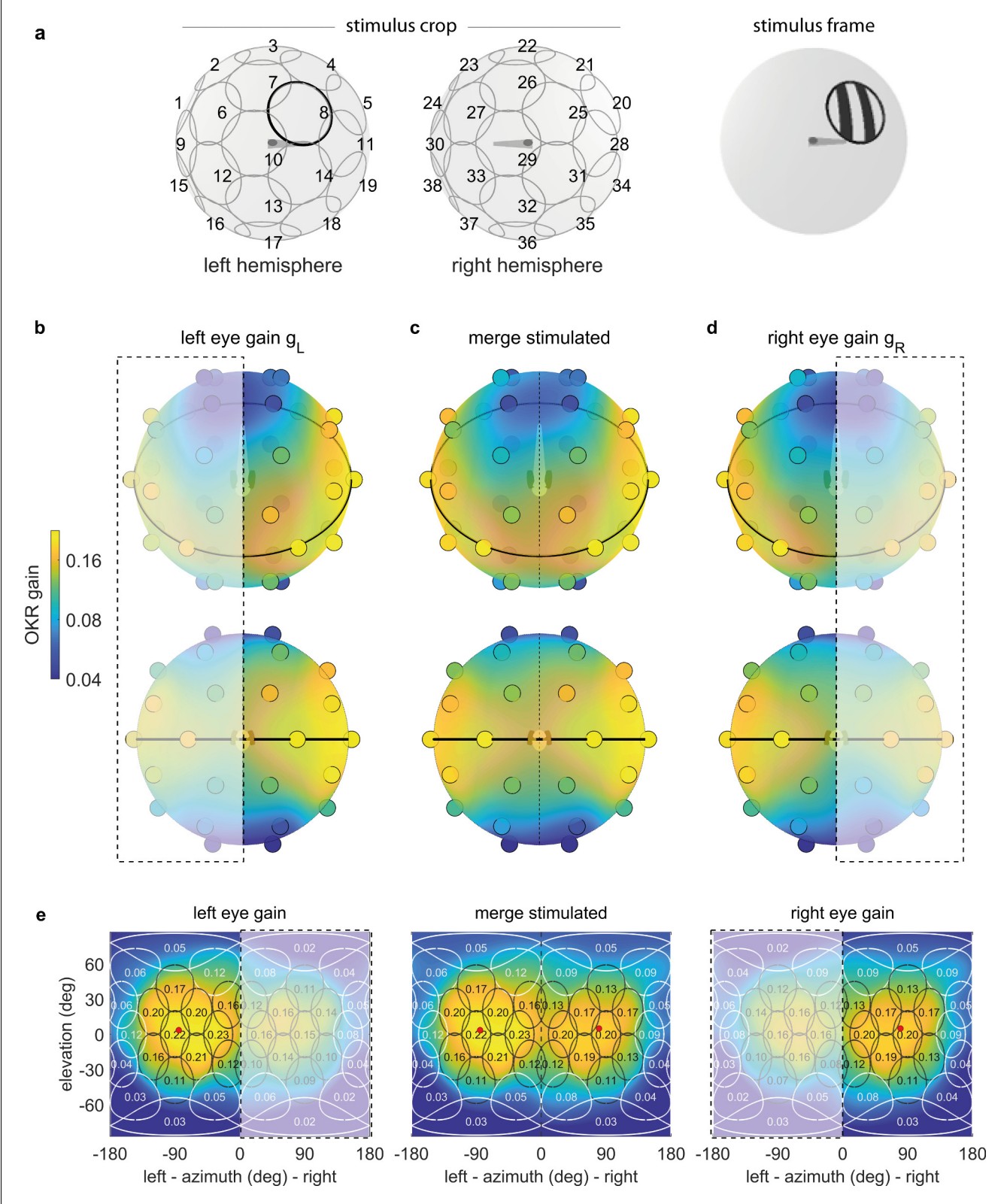

**Figure 3.** OKR gain depends on stimulus location. (a) The stimulus is cropped to a disk-shaped area 40 degrees in diameter, centred on one of 38 nearly equidistant locations (*Supplementary file 1B*) across the entire visual field (left), to yield 38 individual stimuli (right). (b–d) Dots reveal the location of stimulus centres D1-D38. Their colour indicates the average OKR gain across individuals and trials, corrected for external asymmetries. Surface colour of the sphere displays the discretely sampled OKR data filtered with a von Mises-Fisher kernel, in a logarithmic colour scale. Top row: *Figure 3 continued on next page*

*Figure 3 continued*

OKR gain of the left eye (**b**), right eye (**d**), and the merged data including only direct stimulation of either eye (**c**), shown from an oblique, rostrodorsal angle. Bottom row: same, but shown directly from the front. OKR gain is significantly higher for lateral stimulus locations and lower across the rest of the visual field. The spatial distribution of OKR gains is well explained by the bimodal sum of two von-Mises Fisher distributions. (**e**) Mercator projections of OKR gain data shown in panels (**b–d**). White and grey outlines indicate the area covered by each stimulus type. Numbers indicate average gain values for stimuli centred on this location. Red dots show mean eye position during stimulation. Dashed outline and white shading on panels (**b, d, e**) indicate indirect stimulation via yoking, that is, stimuli not directly visible to either the left or right eye. Data from n = 7 fish for the original configuration and n = 5 fish for the rotated arena.

The online version of this article includes the following video, source data, and figure supplement(s) for figure 3:

**Source data 1.** Numerical data and graphical elements of *Figure 3b–d*.
**Source data 2.** Numerical data and graphical elements of *Figure 3e*.
**Figure supplement 1.** Gaps in the arena do not bias OKR behaviour.
**Figure supplement 2.** During OKR, the beating field and average eye position are independent of stimulus location.
**Figure supplement 3.** Vertical eye position under upright and upside-down embedding.
**Figure supplement 4.** Yoking indices are biased by reflections within the arena.
**Figure supplement 5.** Across different types of arenas, stimulus reflections affect perceived yoking.
**Figure 3—video 1.** Animation showcasing short samples of all disk stimuli used to study location dependence, as in *Figure 3a*.
https://elifesciences.org/articles/63355#fig3video1

the visual field, and thus in fish-centred coordinates. Unexpectedly, the elevation of highest OKR gains relative to the fish changed from slightly above to slightly below the equator of the visual field when comparing upright to inverted fish (*Figure 4e*, *Figure 4g*): When upright, azimuths and fish-centred elevations of the peaks of the best fits to data were −78.7° and 8.2° for the left eye, as well as 81.8° and 3.1° for the right eye. When inverted, −88.0° and 1.6° for the left eye, as well as 82.8° and −15.5° for the right eye. These numbers were obtained from the gains of those eyes to which any given stimulus was directly visible. The results from *Figure 4e–f* were combined to correct *Figure 3* for external asymmetries (Materials and methods); this is why the best-fit position reported for *Figure 4e* alone differs slightly from that reported above for *Figure 3b–e*. Because the set of visual stimuli presented to inverted fish stemmed from an earlier stimulus protocol with less even sampling of the visual field, a slight scaling of azimuths and elevations is expected. The consistent reduction of the elevation, however, is not. We performed a permutation test in which embedding-direction labels were randomly swapped while stimulus-location labels were maintained, and the Gaussian-type fit to data was then repeated on each permuted dataset. This test confirmed that fish preferred upward (in the environmental reference frame) rather than dorsalward elevations (p<0.05, *Source code 1E*, *Figure 4—figure supplement 3*). While the fit peaks capture its centres of mass, the apparent maxima of the OKR spatial distribution are even further apart, with their elevation flipping signs from the dorsal to the ventral hemisphere (shown in *Figure 4e, g*).

Adjustment by the fish of its vertical resting eye position between the upright and inverted body positions would have been a simple potential explanation for this result. However, time-lapse frontal microscopy images (Materials and methods) ruled this out, since for both upside-up and upside-down embedding the eyes were inclined by an average of about four degrees towards the dorsum (3.5 ± 1.0° for the left eye, 4.9 ± 0.8° for the right eye, mean ± s.e.m.,

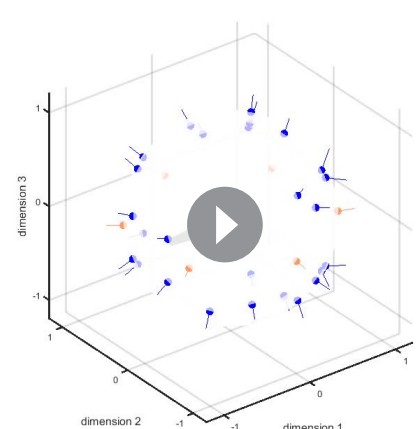

**Video 1.** Repulsion-based algorithm to numerically distribute stimulus centres across a sphere surface (*Source code 1B*), with gradual convergence on a logarithmic timescale (blue to green). A subset of stimulus centres is held on the equator at zero elevation (orange).
https://elifesciences.org/articles/63355#video1

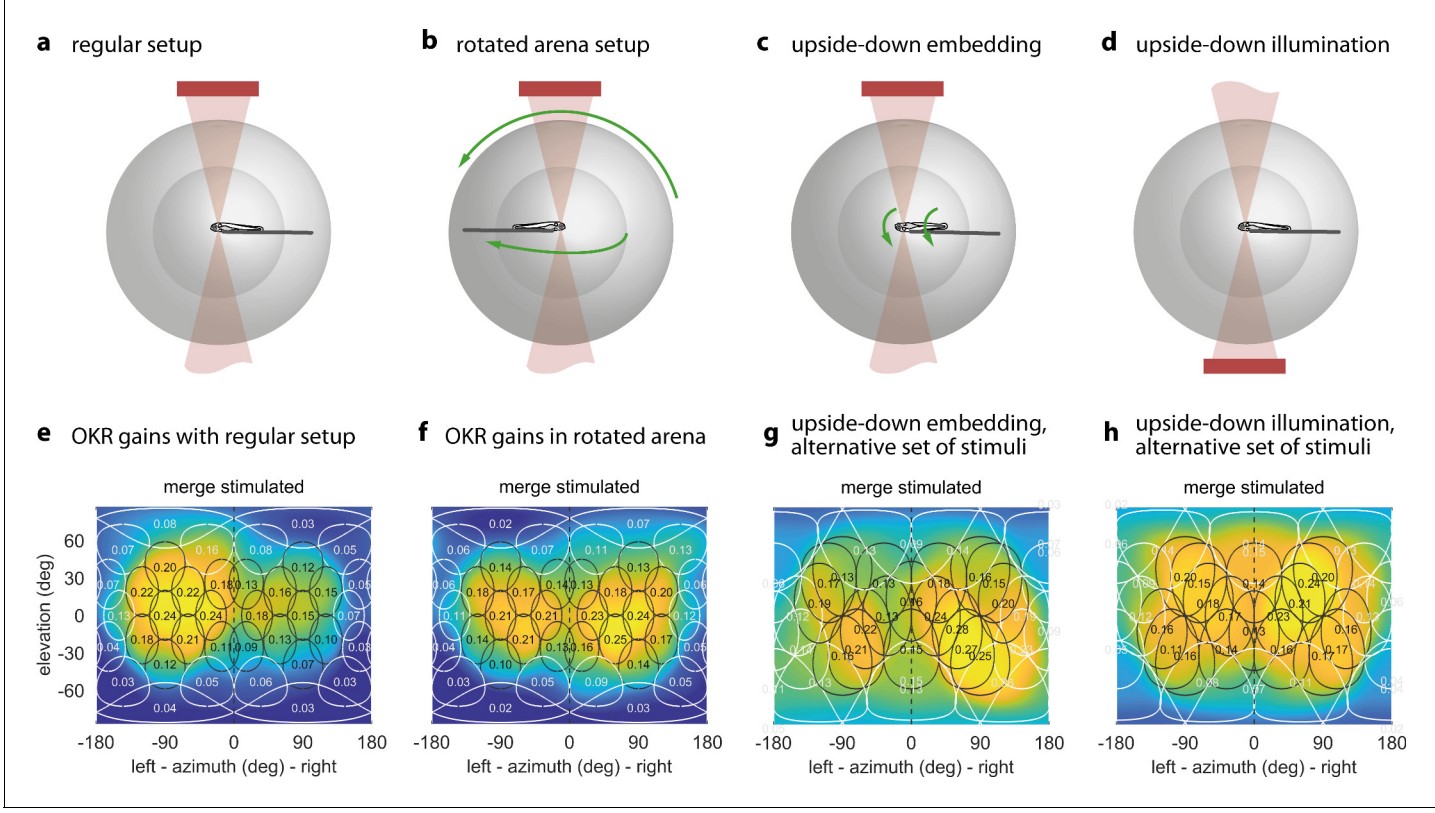

**Figure 4.** The OKR is biased towards upper environmental elevations irrespective of fish orientation. (**a**) Regular arena setup. (**b**) Arena can be tilted 180 degrees so front and rear, upper and lower LED positions are swapped. The bulb holder moves accordingly, so from the perspective of the fish, left and right, upper and lower LEDs are swapped. (**c**) Upside-down embedding setup. (**d**) Setup with inverted optical path, including illumination. (**e–h**) Results in body-centred coordinates, where positive elevations refer to dorsal positions, for the four setups shown in (**a–d**). As in *Figure 3b–e*, colour indicates the discretely sampled OKR data filtered with a von Mises-Fisher kernel and follows a logarithmic colour scale. (**g,h**) Experiments with presentation of a less regularly distributed set of stimuli, cropped to disks of 64 degrees polar angle instead of the 40 degrees used in (**a,b,e,f**). (**g**) Fish embedded upside-down exhibit a slight preference for stimuli below the body-centred equator, that is, positions slightly ventral to their body axis. (**h**) Fish embedded upright, as in (**a**). To account for environmental asymmetries such as arena anisotropies, we combined the data underlying (**e**) and (**f**) to obtain *Figure 3b–e* (see Materials and methods). Data from (**e**) n = 7, (**f**) n = 5, (**g**) n = 3, (**h**) n = 10 fish.

The online version of this article includes the following source data and figure supplement(s) for figure 4:

**Source data 1.** Numerical data and graphical elements of *Figure 4e*.
**Source data 2.** Numerical data and graphical elements of *Figure 4f*.
**Source data 3.** Numerical data and graphical elements of *Figure 4g*.
**Source data 4.** Numerical data and graphical elements of *Figure 4h*.
**Figure supplement 1** Asymmetries between left and right eye are strongly affected by the environment.
**Figure supplement 2.** OKR gain maps for individual larvae.
**Figure supplement 3.** Elevation of stimuli evoking strongest OKR remains upward regardless of embedding direction.

*Figure 3—figure supplement 3*). We also tested the influence of camera and infrared light (840 nm) positions (*Figure 4d*) – which in either case should have been invisible to the fish (*Shcherbakov et al., 2013*) – and found that they could indeed not explain the observed differences (*Figure 4h*). In summary, the body-centred preferred location only flipped from slightly dorsal to slightly ventral in upside-down embedded fish (*Figure 4g*), and thus remained virtually unchanged in environmental coordinates across all control experiments.

Eye-, head-, or body-centred reference frames are therefore not sufficient to fully explain the observed optokinetic stimulus location preferences. Instead, the stimulus location preference is additionally related to the environmental reference frame, suggesting that the behavioural relevance of our motion stimuli depends on their environmental elevation positions.

## Yoking of the non-stimulated eye

Almost all stimuli were presented monocularly – that is, in a position visible to only one of the two laterally located eyes. Without exception, zebrafish larvae responded with yoked movements of both the stimulated and unstimulated eye. To rule out reflections of stimuli within the arena, we performed a series of experiments in which the unstimulated side of the glass bulb had been covered with a matte, black sheet of plastic. Reflections on the glass-air interface would otherwise cause monocular stimuli (that should only be visible to the ipsilateral eye) to also be seen by the contralateral eye. Yoking indices (YI) were significantly different between the regular monocular setup (YI ≈ 0.2) and the control setup (YI ≈ 0.7) containing the black surface on the side of the unstimulated eye, confirming that yoking indices had been affected by reflections (*Figure 3—figure supplement 4*, an index of 1 indicating completely monocular eye movements, an index of 0 perfectly conjugate eye movements/yoking). This suggests that previously reported yoking partially depends on reflections of the stimulus pattern at the glass-to-air or water-to-air interface in our spherical setup and other commonly used stimulus arenas. We performed additional control experiments using a previously described setup (*Dehmelt et al., 2018*) with four flat LCD screens for stimulus presentation in a different room. In these experiments, stimuli (*Supplementary file 1C*) were presented monocularly or binocularly, and the unstimulated eye was either (i) stimulated with a stationary grating (*Figure 3—figure supplement 5a–b*), (ii) shielded with a blank, white shield placed directly in front of the displays (*Figure 3—figure supplement 5c–d*), or (iii) shielded with a matte, black sheet of aluminium foil placed inside the petri dish (control for possible reflections on the Petri dish wall) (*Figure 3—figure supplement 5e–f*). This experiment showed that yoking was much reduced (YI ≈ 0.3) if the non-stimulated eye saw a stationary grating (i) instead of the white or black shields (ii-iii, YI ≈ 0.1) or a binocular motion stimulus (YI ≈ 0) (*Figure 3—figure supplement 5g–h*, p<0.05).

## Spatial asymmetries

As a few previous studies suggested left-right asymmetries in zebrafish visuomotor processing and behaviour other than OKR (*Andrew et al., 2009*; *Watkins et al., 2004*, *Sovrano and Andrew, 2006*) we computed an asymmetry index $B$ (Materials and methods) to reveal whether zebrafish OKR is lateralised in individuals or across the population. We did not observe a general asymmetry between the response of the left and right eyes. Rather, our data is consistent with three distinct sources of asymmetry: individual bias towards one eye, shared bias across individuals, and asymmetries induced by the environment (including the experimental setup and stimulus arena). Through multivariate linear regression, we fit a linear model of asymmetries to our data (Materials and methods), which combined data from fish embedded upside-up (*Figure 4e*), upside-down (*Figure 4g*) and data obtained with the arena rotated relative to the fish (*Figure 4f*), and included whole-field and hemispheric stimuli (*Supplementary file 1D*). Regression coefficients for external causes of asymmetry were similar to those for biological causes (*Figure 4—figure supplement 1*), and individual biases from fish to fish were broadly and symmetrically distributed from left to right (mean coefficient $3.7 \cdot 10^{-4} \pm 120.0 \cdot 10^{-4}$ st. dev., n = 15), so that no evidence was found for a strong and consistent lateralisation of OKR behaviour across animals (*Figure 4—figure supplement 2*).

Our results show that the OKR behaviour is mostly symmetric across both eyes, with individual fish oftentimes having a dominant eye due to seemingly random bias for one eye (lateralisation) across fish. Some of the observed asymmetries are consistent with external factors. Therefore, the OKR gains presented in *Figure 3* have been corrected to present only biologically meaningful differences (Materials and methods).

## Spatial frequency tuning of the optokinetic response is similar across visual field locations

We investigated the spatial frequency tuning of OKR behaviour across visual field positions by presenting seven different spatial frequencies of the basic stimulus, each cropped into a planar angle of 40 degrees, at different visual field locations (*Figure 5a*). We held the temporal frequency constant, therefore stimulus velocity decreased whenever spatial frequency increased. These seven disk-shaped stimuli were presented while centred on one of 6 possible locations in different parts of the visual field (*Figure 5c*), with three locations on each hemisphere: one near the location of highest OKR gain as determined in our experiments on position dependence, one in a nasal location, and

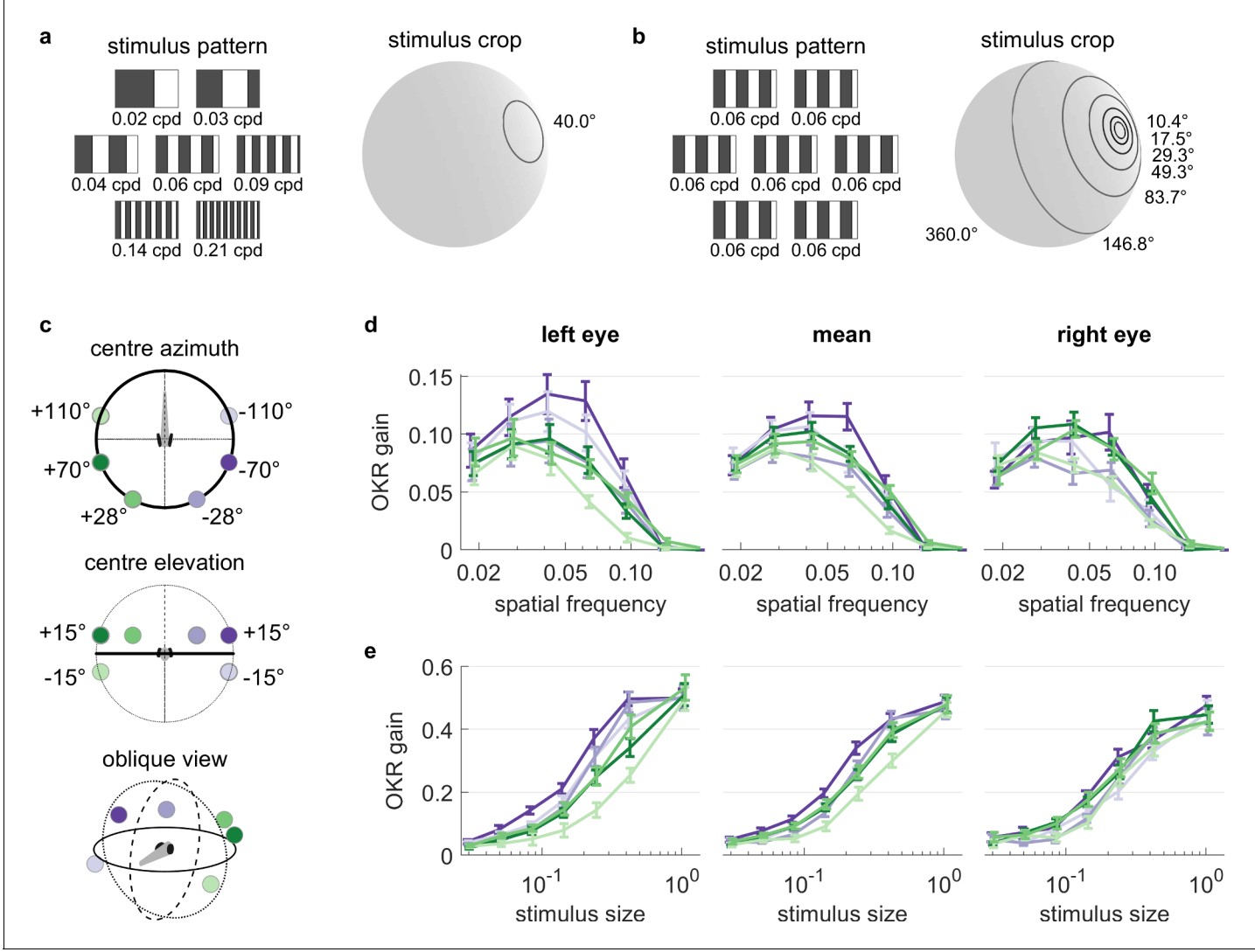

**Figure 5.** OKR tuning to spatial frequency is similar across different visual field locations. (a) Patterns with seven different frequencies were cropped to disks of a single size. These disks were placed in six different locations for a total of 42 stimuli. cpd: cycles per degree. (b) Patterns with identical spatial frequencies were cropped to disks of seven different sizes. These disks were also placed in six different locations for another set of 42 stimuli. Degrees indicate planar angles subtended by the stimulus outline, so 360° correspond to whole-field stimulation. (a, b) Displaying the entire actual pattern at the size of this figure would make the individual bars hard to distinguish. We thus only show a zoomed-in version of the patterns in which 45 out of 360 degrees azimuth are shown. (c) Coloured dots indicate the six locations on which stimuli from a and b were centred, shown from above (top), from front (middle), and from an oblique angle (bottom). (d) OKR gain is unimodally tuned to a wide range of spatial frequency (measured in cycles per degree). (e) OKR gain increases sigmoidally as the area covered by the visual stimulus increases logarithmically (a stimulus size of 1 corresponds to 100% of the spherical surface). (d–e) Colours correspond to the location of stimulus centres shown in (c). There is no consistent dependence on stimulus location of either frequency tuning or size tuning. Error bars show standard error of the mean. Data from n = 7 fish for frequency dependence and another n = 7 fish for size dependence.

The online version of this article includes the following video, source data, and figure supplement(s) for figure 5:

**Source data 1.** Numerical data and graphical elements of *Figure 5*.

**Figure supplement 1.** Magnitude and phase shift of eye movements at different frequencies resemble those previously observed for zebrafish OKR.

**Figure supplement 2.** OKR gain of individual larvae, for different stimulus frequencies.

**Figure supplement 3.** OKR gain of individual larvae, for different stimulus sizes.

**Figure 5—video 1.** Animation showcasing short samples of all disk stimuli used to study frequency dependence, as in *Figure 5a*.
https://elifesciences.org/articles/63355#fig5video1

**Figure 5—video 2.** Animation showcasing short samples of all disk stimuli used to study size dependence, as in *Figure 5b*.
https://elifesciences.org/articles/63355#fig5video2

one in a lower temporal location. In total, we thus presented 42 distinct types of stimuli (*Supplementary file 1F*, *Figure 5—video 1*). For each stimulus location and eye, the highest OKR gain was observed at a spatial frequency of 0.03 to 0.05 cycles/degree (*Figure 5d*, *Figure 5—figure supplement 2*). We did not observe any strong modulation of frequency dependence by stimulus location.

## Size dependence of the optokinetic response

It is unclear to what extent small stimuli are effective in driving OKR. We therefore employed a stimulus protocol with seven OKR stimuli presented in differently sized areas on the sphere (*Figure 5b*). Spatial and temporal frequencies were not altered, so bars appeared with the same width and velocity profile in all cases. These seven disk-shaped stimuli were presented while centred on one of 6 possible locations, identical to those used to study frequency dependence (*Figure 5c*), again yielding 42 unique stimuli (*Supplementary file 1G*, *Figure 5—video 2*). Stimulus area size was chosen at logarithmic intervals, ranging from stimuli almost as small as the spatial resolution of the zebrafish retina, to stimuli covering the entire arena. Throughout this paper, the term 'stimulus size' refers to the fractional area of the sphere surrounding the fish, in which the moving grating stimuli was presented. For instance, with a solid angle of 180°, the stimulus size is 50% and covers half of the surrounding space (*Figure 5b*). In line with many other psychophysical processes, OKR gain increased sigmoidally with the logarithm of stimulus size (*Figure 5e*, *Figure 5—figure supplement 2*). Weak OKR behaviour was already observable in response to very small stimuli of 0.8% (solid angle or 'stimulus diameter' of 10.4°), and reached half-maximum performance at a stimulus size of roughly 25% (solid angle: 120°). As was the case for spatial frequency dependence, we did not observe any strong modulation of size dependence by the tested stimulus locations, although OKR gains of the left eye appeared more dependent on stimulus location than those of the right eye. Specifically, the left eye responded more strongly to ipsilateral than contralateral stimuli (ANOVA, $p < 0.05$), whereas the right eye did not (ANOVA, $p > 0.05$).

## Optokinetic response gain covaries with retinal density of long-wave sensitive photoreceptors

We hypothesised that the non-uniform distribution of the OKR gain across the visual field is related to the surface density of photoreceptors and investigated this using data from a recent study (*Zimmermann et al., 2018*; *Chouinard-Thuly et al., 2017*) on photoreceptor densities in explanted eye cups of 7- to 8-day-old zebrafish larvae. As shown in *Figure 6b*, ultraviolet receptor density exhibits a clear peak in the upper frontal part of the visual field, whereas red, green, and blue receptors (*Figure 6a*) are most concentrated across a wider region near the intersection of the equator and lateral meridian, with a bias to the upper visual field (in body coordinates). For comparison, density maps in retinal coordinates, not body coordinates, are shown in *Figure 6—figure supplement 1*, and *Figure 6c* shows total photoreceptor density across all types. To register our OKR gain data onto the photoreceptor density maps, we took the average eye position into account, which was located horizontally at $-84.8 \pm 6.2$ degrees azimuth for the left and 80.1 ± 6.5 deg for the right eye (mean ±st.dev., n = 7 fish), and vertically at 3.5 ± 3.2 degrees elevation for the left and 4.9 ± 2.7 deg for the right eye (n = 10 fish). For green, blue, and especially red receptors, the visual space location in which OKR gain is maximal (cf, *Figure 3e*) coincides with a retinal region of near-maximum photoreceptor density (white ring in *Figure 6*). For ultraviolet receptors (*Figure 6b*), there is no strong correlation between photoreceptor density and OKR gain.

## Discussion

We used a spherical visual display to systematically investigate visual space anisotropies of the zebrafish optokinetic response. We show that animals react most strongly to stimuli located laterally and near the equator of their visual space. Across individuals, the OKR appears to be symmetric between both eyes, although individual animals oftentimes have a dominant eye. For small stimuli, the OKR gain depends on the size of the area in which the stimulus was presented in a logarithmic fashion. OKR to our mostly yellow stimuli was tuned to the higher spatial densities of red, green, and blue photoreceptors in the central retina. In addition, further processing appears to affect the

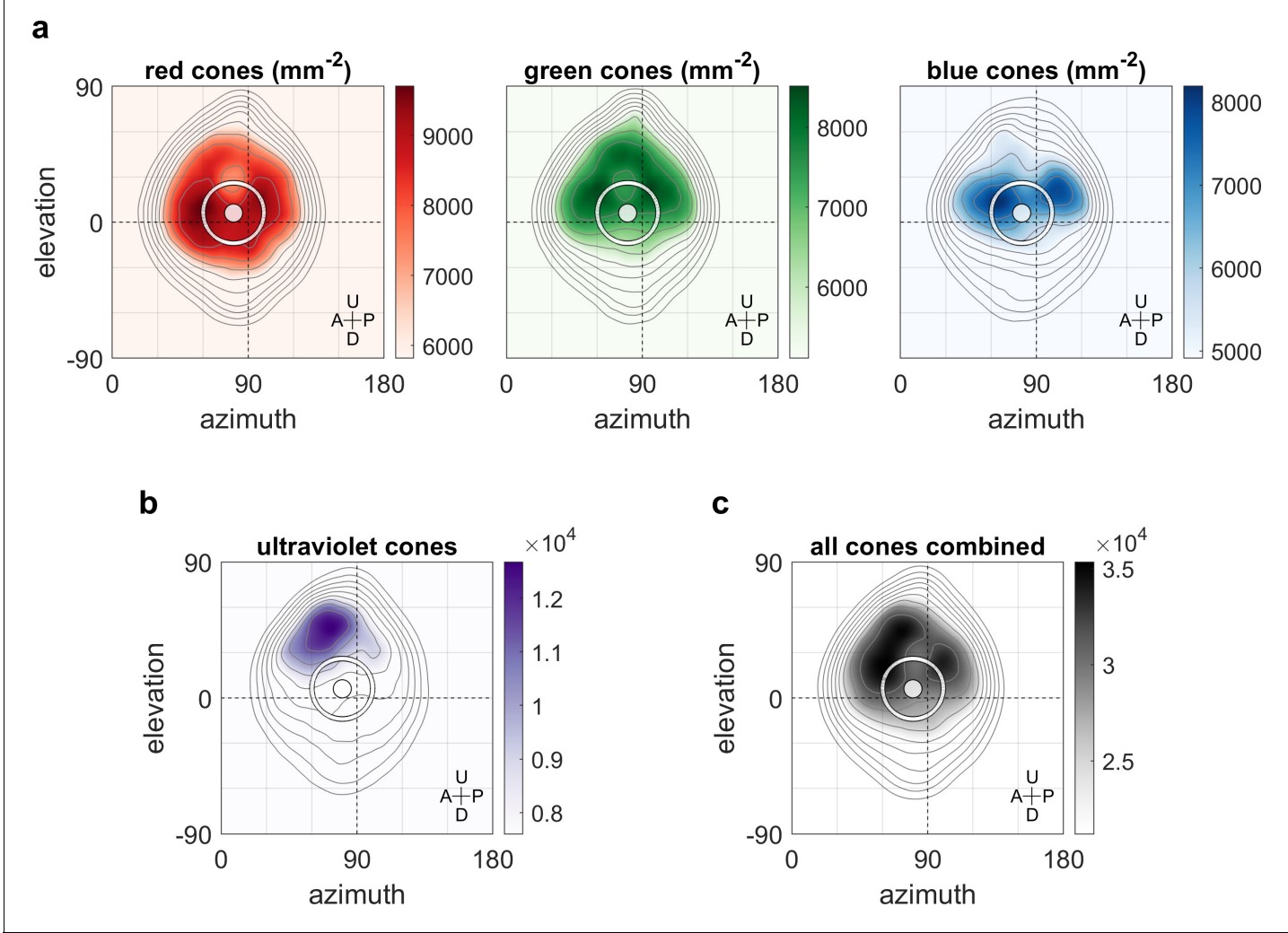

**Figure 6.** Maximum OKR gain is consistent with high photoreceptor densities in the retina. Contour lines show retinal photoreceptor density determined by optical measurements of explanted eye cups of 7–8 dpf zebrafish larvae, at increments of 10% of maximum density. Data shown in visual space coordinates relative to the body axis, that is, 90° azimuth and 0° elevation corresponds to a perfectly lateral direction. To highlight densely covered regions, densities from half-maximum to maximum are additionally shown in shades of colour. Solid white circles indicate the location of maximum OKR gain inferred from experiments of type D in 5-7dpf larvae (**Figure 3**). White outlines indicate the area that would be covered by a 40° disk-shaped stimulus centred on this location when the eye is in its resting position. As the eyes move within their beating field during OKR, the actual, non-stationary retinal coverage extends further rostrally and caudally. For (**a**) red, green, and blue photoreceptors, high densities coincide with high OKR gains. (**b**) For ultraviolet receptors, there is no clear relationship to the OKR gain. (**c**) For reference, the summed total density of all receptor types combined. We did not observe a significant shift in the position-dependence of maximum OKR gain between groups of larvae at 5, 6, or 7 dpf of age, consistent with the notion that retinal development is far advanced and the circuits governing OKR behaviour are stable at this developmental stage. U: Up, D: Down, A: Anterior, P: Posterior.

The online version of this article includes the following source data and figure supplement(s) for figure 6:

**Source data 1.** Numerical data and graphical elements of **Figure 6**.
**Figure supplement 1.** Maximum OKR gain compared to photoreceptor densities in retinal coordinates.

preferred OKR stimulus location, as suggested by the experiments in upside-down embedded animals. This could include extra-retinal processing, or hitherto unknown projections to the retina itself.

The spherical arena introduced here covers a large proportion of the surround and therefore lends itself to many other investigations of zebrafish, and of other species with limited visual acuity. In comparison to other feasible technical solutions, such as video projection setups, our spherical LED array stimulus setup provides homogeneous light and contrast across the entire stimulation area. Thereby, stimulus design is much easier because stimulus warping and conditioning becomes

unnecessary. When combined with calcium imaging in a scanning microscope, the use of LED arrays provides the additional advantage that the visual stimulus can be controlled with high temporal precision, fast enough to interlace visual stimuli and line scans.

Despite the common notion that OKR is a whole-field gaze stabilisation behaviour, our results show that the OKR can be driven effectively by moving stimuli that cover only small parts of the spherical surface (e.g. a half-maximum OKR gain is observed for a stimulus that covers 25% of the spherical surface). In humans, small-field stimuli are effective in driving OKR as well, and the optokinetic drive and motion percept varies for central and peripheral retinal locations, with central stimulus locations eliciting stronger optokinetic drive (*Howard and Ohmi, 1984*; *Brandt et al., 1973*). Our experiment on spatial frequency dependence further demonstrates that the spatial frequency tuning of zebrafish OKR is similar across retinal locations. Since photoreceptors are not equally distributed across the retina, this result suggests that photoreceptor density is not the limiting factor for OKR performance in this frequency range.

Although for OKR and prey capture, animals are thought to respond to stimuli on the left and the right side alike, a few previous reports suggested that the zebrafish visual system is lateralised with the left eye preferentially assessing novel stimuli, while the right eye being associated with decisions to respond (*Sovrano and Andrew, 2006*; *Miklósi and Andrew, 1999*). Given the frequent occurrence of lateralised brain structures and behaviours in the animal kingdom (*Bisazza et al., 1998*; *Güntürkün et al., 2020*), we therefore investigated whether there are consistent behavioural asymmetries for the OKR. We observed almost no consistent, inter-individual asymmetries in OKR between the left and right hemispheres of the visual field, other than those induced by external conditions. Individual fish, however, show a wide and continuous range of biases towards either hemisphere.

We measured OKR gain in larvae at 5–7 days post fertilisation (dpf) of age, whereas our data on photoreceptor densities corresponds to slightly older, 7–8 dpf larvae. Owing to their rapid development, zebrafish undergo noticeable morphological changes on this timescale, but the zebrafish retina itself is known to be well developed by five dpf (*Avanesov and Malicki, 2010*) and stable OKR behaviour is exhibited from then on. Crucially, we did not observe a salient age-dependent spatial shift of maximum OKR gain between our 5 dpf and 7 dpf larvae (data not shown).

The qualitative match between red cone retinal photoreceptor densities and the stimulus position driving the highest OKR gains may provide a mechanistic bottom-up explanation of the gradual differences associated with OKR. The correspondence of red photoreceptor density with the visual field map of OKR gain is consistent with the fact that our LEDs emit light at 568 nm peak power, which should have activated the red cones most. Our data is also in agreement with observations in other species, that the OKR drive is strongest when the moving stimulus covers the central visual field (*Murasugi and Howard, 1989*; *Shimizu et al., 2010*; *Howard and Ohmi, 1984*). In a simplistic, additive view of visual processing, increased numbers of receptors would be triggered by incident light, gradually leading to stronger activation of retinal ganglion cells and downstream circuits, eventually driving extraocular eye muscles towards higher amplitudes. Instead, or in addition, the increased resolution offered by denser distributions of photoreceptors could help reduce sensory uncertainty (and increase visual acuity). It is unclear however, how more uncertainty would lead to consistently lower OKR gains instead of a repeated switching between periods of higher and lower gains, or between OKR and other behaviours. If sensory uncertainty were indeed crucial to OKR tuning, presenting blurred or otherwise deteriorated stimuli should reduce OKR gain in disfavoured locations more strongly than those in favoured locations. It is also possible that correlations between OKR gain and cone photoreceptor density are entirely coincidental, as our spatial frequency tuning results for different stimulus locations had implied. Genetic zebrafish variants with altered photoreceptor distributions would thus be a valuable tool for further studies. Recent studies suggest that melanopsin-expressing ganglion cells also contribute to form vision (*Allen et al., 2019*; *Lucas et al., 2020*), and melanopsins as well as a plethora of other opsins are expressed in the zebrafish retina and brain (*Davies et al., 2015*; *Fontinha et al., 2021*). It is therefore possible that photoreception via opsin-expressing neurons in the inner retina contributes to zebrafish OKR as well.

The pronounced increase in OKR gain for nearly lateral stimulus locations raises questions regarding the top-down behavioural significance of these directions in the natural habitat of larval zebrafish. While reduced OKR gains near the limits of the visual field might be expected, we show that gains are also reduced in the frontal binocular area, as well as in upper and lower visual field

locations. Interestingly, when animals were mounted upside-down, they still prefer stimulus locations just above the equator of the environment. This result cannot be explained by shifted resting vertical eye positions in the inverted animal, which we have measured. Instead, it could potentially be explained by multimodal integration, where body orientation appears to influence the preferred OKR stimulus locations via the vestibular system (*Lafortune et al., 1990*; *Pettorossi et al., 2011*; *Zolotilina et al., 1995*).

Furthermore, it seems possible that the unequal distribution of OKR gains across the visual field is related to the optic flow statistics that naturally occur in the habitats of larval zebrafish (*Zimmermann et al., 2018*; *Arunachalam et al., 2013*; *Engeszer et al., 2007*; *Parichy, 2015*; *Spence et al., 2008*). For another stabilisation behaviour of zebrafish, the optomotor response (*Orger et al., 2008*), we have recently shown that the underlying circuits prefer stimulus locations in the lower temporal visual field to drive forward optomotor swimming (*Wang et al., 2020*). Therefore, the optokinetic and the optomotor response are preferentially driven by different regions in the visual field, suggesting that they occur in response to different types of optic flow patterns in natural habitats. Both the optokinetic and the optomotor response (OKR, OMR) are thought to be mediated by the pretectum (*Kubo et al., 2014*; *Naumann et al., 2016*), and we therefore hypothesise that circuits mediating OKR and OMR segregate within the pretectum and form neuronal ensembles with mostly different receptive field centre locations. Future studies on pretectal visual feature extraction in the context of naturalistic stimulus statistics are needed to establish a more complete picture of the visual pathways and computations underlying zebrafish OKR, OMR and other visually mediated behaviours.

# Materials and methods

## Key resources table

| Reagent type (species) or resource | Designation | Source or reference | Identifiers | Additional information |
|---|---|---|---|---|
| Genetic reagent (*Danio rerio*) | mitfa$^{s184}$ | https://zfin.org/action/genotype/view/ZDB-GENO-080305-2, (*Scott et al., 2007*) | RRID:ZFIN_ZDB-ALT-070629-2 | Mutant fish line (melanocyte inducing transcription factor a) |
| Software, algorithm | ZebEyeTrack | http://zebeyetrack.org/, (*Dehmelt et al., 2018*) | DOI:10.1038/s41596-018-0002-0 | Modular eye tracking software for zebrafish |
| Software, algorithm | spherical arena code | https://gin.g-node.org/Arrenberg_Lab/spherical_arena, (*Dehmelt et al., 2020*) | DOI:10.12751/g-node.qergnn | Code for stimulus generation, LED position mapping, and arena control |
| Software, algorithm | OpenSCAD | https://www.openscad.org/ | | 3D computer-aided design of arena scaffold |
| Other | LED array | Kingbright Electronic Co. | TA08-81CGKWA | Green light emitting LED tiles, 20 × 20 mm each, peak power at 568 nm |
| Other | hardware controller | *Joesch et al., 2008* | DOI:10.1016/j.cub.2008.02.022 | Circuit board designs and code by Alexander Borst (MPI Neurobiol, Martinsried), Väinö Haikala and Dierk Reiff (University of Freiburg) |

## Animal experiments

Animal experiments were performed in accordance with licenses granted by local government authorities (Regierungspräsidium Tübingen) in accordance with German federal law and Baden-Württemberg state law. Approval of this license followed consultation of both in-house animal welfare officers and an external ethics board appointed by the local government. We used *mitfa-/-* animals (5–7 dpf) for the experiments, because this strain lacks skin pigmentation that could interfere with eye tracking. One report suggests that *mitfa-/-* animals show different optomotor responses than wildtype animals (*Stowers et al., 2017*), so it is possible that optokinetic responses investigated here are also modulated by this genetic background.

## Coordinate systems and conventions

To remain consistent with the conventions adopted to describe stimuli and eye positions in previous publications, we adopted an East-North-Up, or ENU, geographic coordinate system. In this system, all positions are relative to the fish itself, and expressed as azimuth (horizontal angle, with positive values to the right of the fish), elevation (vertical angle, with positive values above the fish), and radius (or distance to the fish). The point directly in front of the fish (at the rostrum) is located at [0°, 0°] azimuth and elevation. Azimuth angles cover the range [−180°, 180°] and elevation angles [−90°, 90°]. Azimuth sign is opposite to the conventional mathematical notation of angles when looking top-down onto the fish. In our data repository, we provide a detailed description of the coordinate systems used, including transformations between Cartesian and geographic coordinate systems (*Dehmelt et al., 2020*, https://gin.g-node.org/Arrenberg_Lab/spherical_arena).

Geographic coordinates can be used to identify stimulus position in at least four distinct reference frames: body-centred, head-centred, retina-centred, or world-centred coordinates. Because the heads of our embedded fish remain aligned with the body axis, we do not distinguish between body- and head-centred coordinates, and instead jointly refer to them as 'fish-centred'. We generally identify stimuli by the coordinates of their centre, so there is a unique stimulus position in both fish- and world-centred coordinates. But because the eyes move during OKR, this stimulus centre will fall not on one but on many different retina-centred positions over time. Therefore, our comparison of the OKR gain map to retinal photoreceptor density distributions is based on the time-averaged eye position during optokinetic stimulation.

## Design of the spherical arena

### Geometric design of the arena

The overall layout of the spherical arena was optimised to contain the near maximum number of LED tiles that can be driven by our hardware controllers (232 out of a possible 240), and arrange them with minimal gaps in between. Also, care was taken to leave sufficient gaps near the top and bottom poles to insert the optical pathway used to illuminate and record fish behaviour. A further eight LED tiles could be included as optional covers for the top and bottom poles, bringing the total number to 240 out of 240 possible. A detailed walkthrough of the mathematical planning is found in our data repository (*Dehmelt et al., 2020*, https://gin.g-node.org/Arrenberg_Lab/spherical_arena).

### Arena elements

The arena consists of a 3D-printed structural scaffold (designed with open-source software Open-SCAD, *Figure 1–source data 1*); green light emitting LED tiles (Kingbright TA08-81CGKWA, 20x20 mm each, peak power at 568 nm) hot-glued to the scaffold and connected by cable to a set of circuit boards with hardware controllers (*Figure 1—figure supplement 1d*); 8x8 individual LEDs contained in each tile (*Figure 1f*); a nearly spherical glass bulb filled with water, into which the immobilised larvae are inserted (*Figure 1—figure supplement 1c*, middle); a metal rotation mount attached to the scaffold 'keel' of the arena (*Figure 1—figure supplement 1c*, right), holding the glass bulb in place and allowing corrections of pitch and roll angles; the optical pathway with an infrared light source to illuminate the fish from below (*Figure 1—figure supplement 1b*), and an infrared-sensitive USB camera for video recording of the transmission image (*Figure 1—figure supplement 1d*). In the assembled arena, the whole-field stimulus H1 (see *Supplementary file 1E*) resulted in a luminance of ($10.94 \pm 0.59 cdm^{-2}$) at the arena centre.

### Electronics and circuit design

To provide hardware control to the LEDs, we used circuit boards designs and C controller code provided by Alexander Borst (MPI of Neurobiology, Martinsried), Väinö Haikala and Dierk Reiff (University of Freiburg) (*Joesch et al., 2008*). The electronic and software architecture of stimulus control has originally been designed by *Reiser and Dickinson, 2008* (also see the available documentation for a different version of the CAD and code at (https://reiserlab.github.io/Modular-LED-Display/)). Any custom circuit board design and code could be substituted for these, and alternative solutions exist, for example, in *Drosophila* vision research (*Suver et al., 2016*). At the front end, these electronics control the $8 \times 8$ LED matrices, which are multiplexed in time to allow control of individual LEDs with just 8 input and eight output pins.

## Optical pathway, illumination, and video recording

A high-power infrared LED was placed outside the stimulus arena and its light diffused by a sheet of milk glass and then guided towards the fish through the top hole of the arena (*Figure 1—figure supplement 1b*, *Figure 1—figure supplement 1d*). Non-absorbed IR light exits through the bottom hole, where it is focused onto an IR-sensitive camera. Between the arena and the proximal lens, a neutral density filter (NE13B, Thorlabs, ND 1.3) was inserted half-way (off-axis) into the optic pathway using an optical filter slider (CFH2/M, Thorlabs, positioned in about 5 cm distance of the camera CCD chip) to improve image contrast (oblique detection). We used the 840 nm, 125 degree IR emitter Roschwege Star-IR840-01-00-00 (procured via Conrad Electronic GmbH as item 491118–62) in custom casing, lenses LB1309 and LB1374, mirror PF20-03-P01 (ThorLabs GmbH), and IR-sensitive camera DMK23U618 (TheImagingSource GmbH). Approximate distances between elements are 14.5 cm (IR source to first lens), 12 cm (first lens to centre of glass bulb), 22 cm (bulb centre to mirror centre), 8.5 cm (mirror centre to second lens), 28.5 cm (second lens to camera objective).

## Fish mounting device

Larvae were mounted inside a custom-built glass bulb (*Figure 1—figure supplement 1c*, middle). Its nearly spherical shape minimises reflection and refraction at the glass surface. It was filled with E3 solution, so there was no liquid-to-air boundary distorting visual stimuli. Through an opening on one side, we inserted a glass rod, on the tip of which we immobilise the larva in agarose gel (see description of the embedding procedure below). The fish was mounted in such a way that the head protruded the tip of the narrow triangular glass stage, which ensured that visual stimuli are virtually unobstructed by the glass triangle on their way to the eyes (*Figure 1—figure supplement 1c*, left). The entire glass structure was held at the centre of the spherical arena by metal parts attached to the arena scaffold itself (*Figure 1—figure supplement 1c*, right). Care was taken to remove air bubbles and completely fill the glass bulb with E3 medium.

## Computer-assisted design and 3D printing

To arrange the square LED tiles across a nearly spherical surface, we 3D-printed a structural scaffold or 'skeleton', consisting of a reinforced prime meridian major circle ('keel') and several lighter minor circles of latitude (*Figure 1g*). Available hardware controllers allow for up to 240 LED matrices in parallel, so we chose the exact size of the scaffold (106.5 mm in diameter) to hold as many of these as possible while minimising gaps in between. As individual LEDs are arranged in a rectangular pattern on each of the flat LED tiles, and stimuli defined by true meridians (arcs from pole to pole, or straight vertical lines in Mercator projection), pixelation of the stimulus is inevitable, and stimulus edges become increasing stair-shaped near the poles. Because of the poor visual acuity of zebrafish larvae (see Materials and methods), this should not affect OKR behaviour. Our design further includes two holes necessary for behavioural recordings and two-photon imaging, located at the North and South poles of the sphere. We placed the largest elements of the structural scaffold behind the zebrafish (*Figure 1—figure supplement 1d*). Given the ~160˚ azimuth coverage per eye in combination with a slight eye convergence at rest, this minimises the loss of useful stimulation area.

We printed all structures out of polylactide (PLA) filament using an Ultimaker two printer (Ultimaker B.V.). Parts were assembled using a hot glue gun.

## Visual field coverage of the arena

We can estimate the fraction of the visual field effectively covered by LEDs based on a projection of LED tiles onto a unit sphere. The area $A$ of a surface segment delimited by the projection of the edges of a single tile onto the sphere centre is given by

$$A(S) = \oint dA = \int_{-\lambda}^{\lambda} dx \int_{-\lambda}^{\lambda} dy \left\| u_\alpha \times u_\beta \right\|$$

where $u_\alpha$ and $u_\beta$ are the Cartesian unit vectors spanning the tile itself and $(\pm\lambda, \pm\lambda)$ is the Cartesian position of the four edges of another rectangle. This smaller rectangle is the straight projection of the sphere segment onto the tile,

$$\lambda = \sin\left(\tan^{-1}(D/2R_S)\right)$$

where $R_S = 106.5 mm$ is the sphere radius and $D = 21 mm$ is the length of the edges of the tile. Summing over the number of tiles included in the arena, the equations above can be used to estimate the total coverage of the sphere by its square LED tiles to around 66.5% of the surface area. Using this strict estimate, the small gaps in between LED arrays are counted as not covered, even though we successfully demonstrated that they are small enough not to affect OKR performance, likely due to the low visual acuity of zebrafish larvae. A more meaningful estimate of coverage must take these results into account, and in fact reveals that stimuli presented with our LEDs effectively cover 85.6% of all possible directions:

The top and bottom holes of the arena accommodating the optic path for motion tracking are approximately limited by the circles of +69 and −69 degrees latitude. The fraction of a spherical surface area in between those circles, covered more or less densely by LEDs, corresponds to

$$F = \frac{A_{sphere} - 2A_{cap}}{A_{sphere}} = 1 - \frac{2A_{cap}}{A_{sphere}} = 1 - (1 - \cos(90^\circ - 69^\circ)) = \cos 21^\circ = 93.36\%$$

However, in addition to the holes at the top and bottom of the arena, the structural scaffold of our stimulus arena contains a wide 'keel' on the prime meridian, caudal to the fish, a narrower rostral keel, also on the prime meridian, as well as several thin structural 'ribs' at various latitudes, each without LEDs. The rostral keel spans about 30 degrees azimuth, reducing maximum coverage to 93.36%*(360-30)/360 = 85.58%. Yet this rostral keel is almost always outside the visual field of the zebrafish larvae and thus likely irrelevant for OKR behaviour. Our control experiments further demonstrate that the narrower rostral keel has no discernible effect on OKR gain. We thus assume that the even thinner structural 'ribs' have little to no effect, either; and the fraction of the visual field of zebrafish effectively covered by our stimulus arena thus exceeds 90%.

## Visual acuity and the spatial resolution of the arena

The spatial resolution of the spherical stimulus arena is not very high, providing space for only about 0.54 LEDs per square degree on average. However, larval zebrafish have poor visual acuity, since each retina of their tiny eyes only contains about 10,000 photoreceptors (*Zimmermann et al., 2018*), corresponding to about 1–2 red-green double cone photoreceptors per square degree (*Haug et al., 2010*; *Tappeiner et al., 2012*). The spatial resolution of our arena will not bias the experimental results unless the stimulus is displayed in the most extreme positions.

## Stimulus design

We designed visual stimuli, transformed them to geographical coordinates, and mapped them onto the physical positions of each individual LED with custom MATLAB software. We have made this code available for free under a Creative Commons NC-BY-SA 4.0 license ( *Source code 1A*). The mapped stimulus was then uploaded to the hardware controllers using custom-built C code originally developed by Väinö Haikala. Each stimulus type was shown twice to each individual. First, all stimulus types were shown in random order, with a different random sequence for each larva; then, the same stimuli were shown a second time, in the opposite order as before.

To investigate OKR gain dependence on stimulus location, we chose to present stimuli centred on 36 different locations distributed nearly equidistantly across the spherical arena, as well as symmetrically distributed between the left and right, upper and lower, front and rear hemispheres (*Figure 3a*). These positions were determined numerically: First, we populated one eighth of the sphere surface by placing one stimulus centre at a fixed location at the intersection of the equator and the most lateral meridian (90 degrees azimuth, 0 degrees elevation), constraining two more stimulus centres to move along this lateral meridian (90 degrees azimuth, initially random positive elevation), constraining yet another stimulus centre to move along the equator (initially random positive azimuth, 0 degrees elevation), and allowing three more stimulus centre to move freely across the surface of this eighth of the sphere (initially random positive azimuth and elevation), for a total of seven positions. Second, we placed additional stimulus centres onto all 29 positions that were mirror-symmetric to the initial 7, with mirror planes placed between the six hemispheres listed above. We then simulated interactions between all 38 stimulus centres akin to electromagnetic repulsion,

until a stable pattern emerged. Resulting coordinate values were rounded for convenience (*Source code 1B*, *Video 1*).

## Embedding procedure

To immobilise fish on the glass tip inside the sphere, we developed a novel embedding method. A cast of the glass triangle (and of the glass rod on which it is mounted) was made by placing it inside a Petri dish, which was then filled with a heated 2% agarose solution. After agarose cooled down and polymerised, agarose within a few millimetres of the tip of the glass triangle was manually removed, before removing the triangle itself. The resulting cast was stored in a refrigerator and then used to hold the glass triangle during all subsequent embedding procedures, limiting the freedom of movement of the larva to be embedded. The triangle was stored separately at room temperature. Before each embedding, we coated the glass triangle with polylysine and dried it overnight in an incubator at 29°C to increase the subsequent adhesion of agarose. We then returned the glass triangle into its cast, and constructed a tight, 2 mm high circular barrier around its tip using pieces of congealed agarose. A larva was picked up with as little water as possible using a glass pipette and very briefly placed inside 1 ml of 1.6% low-melting agarose solution at 37°C. Using the same pipette, the larvae was then transferred onto the glass triangle along with the entire agarose. After the larva had been placed a few millimetres away from the tip of the glass triangle, the orientation of the animal could be manipulated with custom-made platinum wire tools without touching its body, as previously described (*Arrenberg, 2016*). Before the agarose congeals, swimming motions of the animal were exploited to guide it towards the tip and ensure an upright posture. The final position of the fish was chosen as such that its eyes are aligned with the axis of the glass rod, its body is upright without any rotation, and its head protrudes forward from the tip of the glass triangle, maximising the fraction of its field of view unobstructed by glass elements. The agarose was left to congeal, and the Petri dish was filled with in E3 solution. The freshly congealed agarose surrounding the glass triangle was then removed using additional, flattened platinum wire tools, once again separating the glass triangle from the cast. Using the same tools, we finally cut triangular holes into the remaining agarose to completely free both eyes. To ensure free movement of both eyes, we confirmed the presence of large and even optokinetic eye movements using a striped paper drum before the experiment.

We then pick up the glass triangle by the glass rod attached to it, cut off any remaining agarose detritus, and place it inside the E3-filled glass bulb. No air remained in the bulb, and no pieces of detritus were introduced into the bulb, as these would accumulate near the top and bottom of the bulb, respectively, interfering with the optical pathway and thus reduce image quality.

## Vertical eye position

A separate group of 5 dpf and 6 dpf fish were individually embedded in agarose, their eyes freed, and spontaneous eye position filmed at 60 Hz for about 3 min both along the dorsoventral axis and along the rostrocaudal axis simultaneously. Because changes in horizontal eye position affect the apparent shape of the eye when seen from the front, we sampled the recording with 1 Hz to evaluate the horizontal eye trace. The baseline eye position in the dorsoventral view was determined by mean ±0.5 std. Next, vertical eye position was evaluated if and only if the eyes in the dorsoventral view were at their baseline position. The vertical eye position was further corrected by embedding angle relative to the environmental horizon.

## Data analysis

Video images of behaving zebrafish larvae were processed in real time using a precursor of the *ZebEyeTrack* software (*Dehmelt et al., 2018*), available from http://www.zebeyetrack.com/. The resulting traces of angular eye position were combined with analogue output signals from the hardware controllers of the spherical arena to match eye movement to the various stimulus phases. This was achieved using custom-built MATLAB software, which is freely available under a Creative Commons NC-BY-SA 4.0 license ( *Source code 1C).*

Data was then analysed further by detecting and removing saccades and fitting a piece-wise sinusoidal function to the eye position traces. The parameters of the fit were then compared to the parameters of the equally sinusoidally changing angular positions of the stimulus. For each fish, eye,

and stimulus phase, the ratio between the amplitude of the fit to eye position and the amplitude of stimulus position represents one value of the gain of the optokinetic response.

For each interval between two subsequent saccades, or inter-saccade-interval (ISI), the fit function to the eye position data is defined by

$$f(t \in ISI_k) = -c_1 \cos(c_2 t + c_3) + c_{k+3}$$

Here, $t$ are the time stamps of data points falling within the $k$-th ISI, $c_1$, $c_2$, and $c_3$ are the amplitude, frequency and phase shift of oscillation across all ISIs, and $c_{k+3}$ is a different constant offset within each ISI, which corrects for the eye position offsets brought about by each saccade. The best fit value $c_1$ was taken as an approximation of the amplitude $a_E$ of eye movement, $a_E \approx c_1$. The process of cropping saccades from the raw data and fitting a sinusoid to the remaining raw data is demonstrated in *Figure 1—figure supplement 1*.

The OKR gain $g$ is a common measure of visuomotor function. It is defined as the ratio between the amplitude $a_E$ of eye movement and the amplitude $a_S$ of the visual stimulus evoking eye movement,

$$g = \frac{a_E}{a_S} = \frac{c_1}{a_S}$$

In other words, OKR gain indicates the degree to which zebrafish larvae track a given visual stimulus. For each eye, a single gain value per stimulus phase is computed. While a value of 1 would indicate a 'perfect' match between eye movement and stimulus motion, zebrafish larvae at 5 dpf often exhibit much lower OKR gains (*Rinner et al., 2005*). While highest gains are obtained for very slowly moving stimuli, in our experiments, we chose higher stimulus velocities. Although these velocities are only tracked with small gains, the absolute velocities of the eyes are high, which allowed us to collect data with high signal-to-noise levels and reduce the needed recording time.

To rule out asymmetries induced by the arena itself or by its surroundings, we recorded two sets of stimulus-position-dependence data, one with the arena in its original configuration, and another with the arena rotated by 180 degrees (*Figure 4a–b*). Each set contained data from multiple larvae, and with at least two separate presentations of each stimulus position. For each stimulus position, and separately for both sets of data, we computed the median OKR gain across fish and stimulus repetitions. We then averaged between the two datasets, yielding a single OKR gain value per stimulus position. As asymmetries are less crucial when studying stimulus frequency and size (*Figure 5*), we did not repeat those with a rotated arena, and could thus omit the final step of the analysis.

## Von Mises-Fisher fits to data

Based on the assumption that OKR position tuning could be normally distributed with respect to each angle, OKR gain would be approximated by a two-dimensional, circular von Mises-Fisher function centred on the preferred stimulus location. Because the eyes are yoked, the OKR gain of one eye will be high around its own preferred position, as well as around the preferred position of the contralateral eye. To account for this, we fit the sum of two independent von Mises-Fisher functions to our OKR gain data:

$$F(\alpha, \beta) = \frac{C_1 \kappa_1 \exp\left(\kappa_1 \mu_1^T \xi\right)}{2\pi(\exp(\kappa_1) - \exp(-\kappa_1))} + \frac{C_2 \kappa_2 \exp\left(\kappa_2 \mu_2^T \xi\right)}{2\pi(\exp(\kappa_2) - \exp(-\kappa_2))} + C_3$$

Here, $\xi$ is the Cartesian coordinate vector of a point on the sphere surface, and corresponds to the geographic coordinates azimuth $\alpha$ and elevation $\beta$. $\mu_1$ and $\mu_2$ are Cartesian coordinate vectors pointing to the centre of the two distributions, and $\kappa_1$ and $\kappa_2$ express their respective concentrations, or narrowness. The parameters $\mu_j$, $\kappa_j$, the amplitudes $C_1$, $C_2$ and the offset $C_3$ are fit numerically.

*Figure 3* and *Figure 4e–h* show the best von-Mises-Fisher fits to data as coloured sphere surfaces, whereas the individual data points are shown as small coloured circles in *Figure 3*. All results shown in *Figure 3* and *Figure 4e–h* pertain to a single, constant size of cropping disks, and the colour of any point across the sphere surface represents the expected value of OKR gain if another stimulus were shown with a cropping disk centered on this location.

## Permutation test

We tested whether the preferred stimulus positions are indeed 'above' in world-centred coordinates, instead of 'dorsal' in body-centred coordinates. Two groups of fish with different embedding direction were compared (*Figure 4a* vs. *Figure 4c*). Within each group, we pooled data from all individuals, then computed the best von Mises-Fisher fit. Within each group, we then averaged between the two elevations obtained for the left and right hemispheres, to obtain a single measure of elevation per group. Our test statistic was the difference in (body-centred) elevation between the groups. We then permuted the group-identity labels of each fish, thus assigning fish to two new groups. We then computed the test statistic again for each permuted dataset. This was repeated for all 120 possible permutations that maintain the same relative group size as the two experimental groups. The reported p-value is the fraction of permutations resulting in a test statistic greater or equal to that of the original, unpermuted groups (*Figure 4—figure supplement 3*).

## Yoking index, asymmetry, and mathematical modelling

To quantify asymmetries in the gain between left and right, stimulated and unstimulated eyes, we introduce the yoking index

$$Y = \frac{g_L - g_R}{g_L + g_R}$$

Here, $g_L$ and $g_R$ are the OKR gains of the left eye and right eye, measured during the same stimulus phase. Depending on stimulus phase, only the left eye, only the right eye or both eyes may have been stimulated. If the yoking index is positive, the left eye responded more strongly than the right eye; if it is negative, the amplitude of right eye movement was larger. An index of zero indicates 'perfect yoking', that is identical amplitudes for both eyes.

In addition, we define a 'bias' index to capture innate or induced asymmetries between responses to stimuli presented within the left or right hemisphere of the visual field,

$$B = \frac{m_L - m_R}{m_L + m_R}$$

Here, $m_L$ and $m_R$ are the medians of OKR gains after pooling across either all left-side or all right-side stimulus types (D1-D19 and D20-D38, respectively). Several sources of asymmetry contribute to $B$: (1) arena- or environment-related differences in stimulus perception, constant across individuals; (2) a biologically encoded preference for one of the two eyes, constant across individuals; (3) inter-individual differences between the eyes, constant across stimulus phases for each individual; (4) other sources of variability unaccounted for, and approximated as a noise term $\eta$. We hypothesise that the overall asymmetry observed for each larva $k$ is given by a simple linear combination of these contributions,

$$B_k = \varphi b_1 + b_2 + b_{3,k} + \eta$$

The parameter $\varphi$ is 1 for the default arena setup, and –1 during control experiments with a horizontally flipped arena setup. To determine $b_1$, $b_2$ and $b_3$, we fit this system of equations by multivariate linear regression to experimentally observed bias indices. The system is initially underdetermined, as it contains $n + 2$ coefficients for every $n$ fish observed. However, if we assume that individual biases average out across the population, we can determine the population-wide coefficients $b_1$ and $b_2$ by setting aside the individual $b_{3,k}$ for a first regression. To determine how far each individual deviates from the rest of the population, we then substitute their best regression values of $b_1$ and $b_2$ into the full equation, and perform a second regression for the remaining $b_{3,k}$.

## Data repository

The raw recordings as well as pre-processed datasets and programming scripts to generate the figures are available at a GIN repository at https://gin.g-node.org/Arrenberg_Lab/spherical_arena/ (*Dehmelt et al., 2020*).

## Acknowledgements

We thank Väinö Haikala and Dierk F Reiff (University of Freiburg) for sharing their code and design for hardware controllers, Alexander Borst (MPI Neurobiology, Martinsried) for providing the LED panel board design based on original precursory versions designed by Reiser et al. (*Reiser and Dickinson, 2008*), Thomas Nieß (glassblowing workshop, University of Tübingen) and Klaus Vollmer (precision mechanics workshop, University of Tübingen) for technical support, Prudenter-Agas (Hamburg, Germany) for generating glass bulb illustrations, Andre Maia Chagas for help with 3D printing procedures, and Jan Benda (University of Tübingen) for discussions on visual acuity. Funding This work was funded by the Deutsche Forschungsgemeinschaft (DFG) grants EXC307 (CIN – Werner Reichardt Centre for Integrative Neuroscience) and a Human Frontier Science Program (HFSP) Young Investigator Grant RGY0079.

## Additional information

### Funding

| Funder | Grant reference number | Author |
|---|---|---|
| Deutsche Forschungsgemeinschaft | EXC307 | Aristides B Arrenberg |
| Human Frontier Science Program | RGY0079 | Aristides B Arrenberg |

The funders had no role in study design, data collection and interpretation, or the decision to submit the work for publication.

### Author contributions

Florian A Dehmelt, Software, Formal analysis, Supervision, Validation, Visualization, Methodology, Writing - original draft, Writing - review and editing; Rebecca Meier, Software, Formal analysis, Validation, Investigation, Methodology, Writing - original draft; Julian Hinz, Methodology, Writing - original draft, Writing - review and editing; Takeshi Yoshimatsu, Tom Baden, Resources, Writing - review and editing; Clara A Simacek, Ruoyu Huang, Validation, Investigation; Kun Wang, Methodology, Writing - review and editing; Aristides B Arrenberg, Conceptualization, Software, Funding acquisition, Methodology, Writing - original draft, Project administration, Writing - review and editing

### Author ORCIDs

Florian A Dehmelt ⬡ https://orcid.org/0000-0001-6135-4652
Julian Hinz ⬡ https://orcid.org/0000-0002-5432-6263
Tom Baden ⬡ http://orcid.org/0000-0003-2808-4210
Aristides B Arrenberg ⬡ https://orcid.org/0000-0001-8262-7381

### Ethics

Animal experimentation: Animal experiments were performed in accordance with licenses granted by local government authorities (Regierungspräsidium Tübingen) in accordance with German federal law and Baden-Württemberg state law. Approval of this license followed consultation of both in-house animal welfare officers and an external ethics board appointed by the local government.

### Decision letter and Author response

Decision letter https://doi.org/10.7554/eLife.63355.sa1
Author response https://doi.org/10.7554/eLife.63355.sa2

# Additional files

## Supplementary files

• Source code 1. Five separate sets of MATLAB code. (**A**) Code to design visual stimuli, convert them to geographic coordinates, and map them onto the actual position of individual LEDs. (**B**) Code to numerically identify a distribution of nearly equidistant stimulus centres that is symmetric between the left and right, upper and lower, as well as front and rear hemispheres. (**C**) Code to read raw eye traces, identify individual stimulus phases, detect and remove saccades, compute piece-wise fits to cropped and pre-processed raw data, and return OKR gains. (**D**) Code to recreate the results figures from data, especially *Figure 3*, *Figure 5*, *Figure 6*. (**E**) Code to assess the significance of differences between the best fits to data for fish embedded upright or upside-down. This last set of code requires access to the raw data repository.

• Supplementary file 1. Seven supplementary tables. (**A**) The arena cross-section, (**B**) stimulus parameters for position dependence experiments, (**C**) absolute positions of fish and setup elements in different experiments, (**D**) stimulus parameters for control experiments, (**E**) parameters for whole-field and hemisphere stimuli, (**F**) stimulus parameters for frequency dependence experiments, and (**G**) stimulus parameters for size dependence experiments.

• Transparent reporting form

## Data availability

Analysis code, pre-processed data and examples of raw data have been deposited in GIN by G-Node and published (https://doi.org/10.12751/g-node.qergnn).

The following dataset was generated:

| Author(s) | Year | Dataset title | Dataset URL | Database and Identifier |
|---|---|---|---|---|
| Dehmelt FA, Meier R, Hinz J, Yoshimatsu T, Simacek CA, Wang K, Baden T, Arrenberg AB | 2020 | Gaze stabilisation behaviour is anisotropic across visual field locations in zebrafish | https://gin.g-node.org/Arrenberg_Lab/spherical_arena.git | GIN (G-Node Infrastructure), 10.12751/g-node.qergnn |

The following previously published dataset was used:

| Author(s) | Year | Dataset title | Dataset URL | Database and Identifier |
|---|---|---|---|---|
| Zimmermann MJY, Nevala NE, Yoshimatsu T, Osorio D, Nilsson DE, Berens P, Baden T | 2019 | Data from: Zebrafish differentially process colour across visual space to match natural scenes | https://doi.org/10.5061/dryad.5bc8vd7 | Dryad Digital Repository, 10.5061/dryad.5bc8vd7 |

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
