## [Decision Letter]

**Acceptance summary:**

The paper is of interest to people studying visual systems neuroscience. The authors used an impressive new experimental set-up to determine the dependency of optokinetic response tuning to stimulus location, size, and frequency.

**Decision letter after peer review:**

Thank you for submitting your article "Gaze stabilisation behaviour is anisotropic across visual field locations in zebrafish" for consideration by *eLife*. Your article has been reviewed by 2 peer reviewers, and the evaluation has been overseen by a Reviewing Editor and Tirin Moore as the Senior Editor. The following individual involved in review of your submission has agreed to reveal their identity: Andrew D Straw (Reviewer #1).

The reviewers have discussed the reviews with one another and the Reviewing Editor has drafted this decision to help you prepare a revised submission.

We would like to draw your attention to changes in our revision policy that we have made in response to COVID-19 (https://elifesciences.org/articles/57162). Thus the revisions requested below will likely be addressable without any additional lab experiments.

The reviewers and the Reviewing Editor agreed that the design of the set-up presents a technical innovation that will be highly useful for future studies. There is clear disagreement on the level of conceptual insight.

Based on the technical advancement we agreed on a major revision of the manuscript. However, the resubmission should not be handed in as a research article, but as a Tools and Resources article.

Please address our major concerns listed below.

1) Statistical analysis. For several results only means are shown, and often there does not appear to be an indication of the variance between individuals. For instance:

– In Figures3 and 4 it would be interesting to know how this varies across animals. One possibility would be to also plot a variance map. Another would be to include the maps for each animal individually as supplementary data.

– In Figure 5 error bars should be shown in d,e, and also perhaps the plots for individual animals as supplementary data.

– Line 123: how variable is the visual field size? The current writing suggests it's exactly 163 degrees in every fish.

– Line 248: What is the variance over animals of 80.3 degrees?

– Line 289-292: What is the variance in these numbers?

– Line 299: please show us what the distribution for the permutation test looks like.

2) The authors argue that "extra-retinal processing" is responsible for the minor shift in the center of the highest gain OKR spot. However, while the effect indeed seems likely to require extra-retinal input, the authors provide no evidence that the processing mediating this should occur outside of the retina. Given other examples of non-visual information being sent to the retina, it is at least plausible that retinal processing is involved.

– Line 79: The evidence for it being "extra-retinal processing" is lacking. The source of the effect is likely extra-retinal but whether the processing happens in the retina or elsewhere is not shown.

– 182: The "determinants" are indeed likely extra-retinal, but this does not mean the processing is.

– Line 428: You have not demonstrated that the processing is extra-retinal, only that it is not occurring in retinal coordinates. There could be a projection into the retina, for example.

3) The authors should consider three coordinate systems explicitly, particularly in the lines 273-303. These would be retinal coordinates, head coordinates and environmental coordinates.

– Line 277: It seems there's also another representation which is missing from your consideration here – a body centric representation. This is distinct from retinal coordinates and distinct from environmental coordinates but is known from primate studies, for example.

– Line 282. Now retina-centered is missing. (The retina can move independently from the head/body).

4) Especially as we consider this manuscript rather an advancement for the "Tools and Resources" section, please consider that the Stowers J et al. 2017 "Virtual Reality for Freely Moving Animals" was a full-surround setup – at least as defined by the standard for the other citations – used successfully on *Drosophila* and, importantly here, fish. You need to be more specific in your wording (e.g. to limit statements to systems based on LEDs capable of producing the desired spectral content) or cite that paper.

5) The statement that the gain of the OKR broadly follow the density of the longer-wavelength sensitive photoreceptors in the retina is entirely ignoring the fact that the fish retina is full of Opsins in all kind of different cells types. This includes different melanopsins, but also VA-opsins (which are likely sensitive in the green), opn5s (UVA), tmt-opsins etc. Evidence from mouse and humans starts to suggest that even in these mammals- which have a highly reduced number of non-rod, non-cone opsins- melanopsin+ RGCs are involved in vision (see Annu Rev Vis Sci. 2020 Sep 15;6:453-468. doi: 10.1146/annurev-vision-030320-041239 for a recent review). This aspect needs to be addressed in their optokinetic response gain covariant analyses.

6) Spectral information on the LEDs needs to be provided (in photons per area and time).

---

## [Author Response]

The reviewers and the Reviewing Editor agreed that the design of the set-up presents a technical innovation that will be highly useful for future studies. There is clear disagreement on the level of conceptual insight.Based on the technical advancement we agreed on a major revision of the manuscript. However, the resubmission should not be handed in as a research article, but as a Tools and Resources article.Please address our major concerns listed below.1) Statistical analysis. For several results only means are shown, and often there does not appear to be an indication of the variance between individuals. For instance:– In Figures3 and 4 it would be interesting to know how this varies across animals. One possibility would be to also plot a variance map. Another would be to include the maps for each animal individually as supplementary data.

We chose the latter option suggested by the reviewer and have now included the OKR gain maps for each individual animal as our new Figure 4—figure supplement 2. Inspection of single-fish data led us to conclude that data from one individual from one of the control groups (fish no. 1 from the rotated arena group) had been corrupted. The seemingly arbitrary spatial arrangement of OKR gains likely indicates a mistake by the experimenter, such as misidentifying the stimulus presentation order, and cannot be corrected post-hoc. For maximum transparency, we still show the results from this one fish in the first panel of Figure 4—figure supplement 2b, but we no longer include them in further analyses.

In the process of revisiting our OKR gain maps, we realised that while our von-MisesFisher (vMF) fits accurately identify the centres of mass of the spatial distribution of OKR gains, their appearance tends to mislead readers about the details of the OKR gain distribution away from these points. Most importantly, frontal stimuli (azimuth and elevation 0 degrees) still evoke intermediate-magnitude OKR, but were previously shown in similar colour as polar stimuli (elevation near +/- 90 degrees), which evoke much weaker OKR. To avoid misleading viewers, we now replaced our previous heat maps (showing only the best vMF fit to data) with more informative heat maps (showing the raw OKR gain data filtered with a vMF kernel and using a logarithmic colour map). Figure 3, Figure 4 and Figure 4—figure supplement 2 have been changed accordingly.

– In Figure 5 error bars should be shown in d,e, and also perhaps the plots for individual animals as supplementary data.

Agreed.

Action taken: We have now included error bars in Figure 5 D-E. We have also created the new Figure 5—figure supplement 2 (stimulus frequency) and Figure 5—figure supplement 3 (stimulus area) showing the data from individual animals.

– Line 123: how variable is the visual field size? The current writing suggests it's exactly 163 degrees in every fish.

While there is some evidence that visual field size increases early development, little is known about variability between individuals. The value of 163 degrees cited refers to that reported by Easter and Nicola, 1996 Dev Biol, for 4 dpf larvae. The same authors report a smaller visual field for younger fish, namely 139 degrees at 3 dpf. As all our fish were older than 4 dpf, we assume their visual fields to cover at least 163 degrees.

Action taken: To avoid giving the false impression of an exact known value across fish, we have now rephrased the sentence. It reads:

“Zebrafish are lateral-eyed animals and have a large visual field, which increases during early development and in individual 4day old larvae has been reported to cover about 163° per eye (17), though little is known about interindividual variability.”

– Line 248: What is the variance over animals of 80.3 degrees?– Line 289-292: What is the variance in these numbers?

Because each individual fish can only be tested for a few hours at a time, and because their rapid development precludes repeated testing on subsequent days, each stimulus type could only be shown twice to each animal (that is, for 2x100 seconds). We thus obtain only two gain values per stimulus per fish, so individual-fish data is inherently variable, and only part of this variability represents genuine biological interindividual variability. We thus made the conscious choice not to report “variance over animals” in this context, and instead pooled the data across fish before further analysis. After pooling and averaging across fish, results from the main experimental group were further corrected with those from one of our control groups (“rotated arena”). Only then did we apply our fitting procedure, which was designed for this specific purpose, and may not be reliable for the much noisier and uncorrected individual-fish data. To provide the reviewer with the requested numbers we nonetheless applied the procedure, and provide the resulting best-fit coordinates for the reviewers’ discretion, but caution against their reliability.

Action taken: No changes to the manuscript itself, but we computed best fits to individual-fish data to better inform the reviewers (see Author response table 1 and Author response table 2 listing angles in degrees, and in a body-centred reference frame):

**Author response table 1. resptable1:** 

regular arena	azimuth 1	elevation 1	azimuth 2	elevation 2
fish no. 1	-77.6	7.5	70.7	9.2
fish no. 2	-78.7	6.4	52.9	13.2
fish no. 3	-76.5	8.2	64.7	8.0
fish no. 4	-78.7	6.4	52.9	13.2
fish no. 5	-77.9	5.3	92.6	0.0
fish no. 6	-90.7	3.3	84.8	-2.9
fish no. 7	-83.2	9.9	91.9	-3.3

**Author response table 2. resptable2:** 

rotated arena	azimuth 1	elevation 1	azimuth 2	elevation 2
fish no. 1*	-27.4	34.2	141.5	-39.6
fish no. 2	-64.9	10.3	71.7	-9.6
fish no. 3	-76.3	0.9	86.7	4.5
fish no. 4	-97.9	7.6	73.1	-6.6
fish no. 5	-93.3	-1.9	76.9	4.8

*fish excluded from all analyses as explained above and in manuscript

– Line 299: please show us what the distribution for the permutation test looks like.

Yes, we agree that this information is helpful. The reviewer’s request also prompted us to notice that a detailed verbal description of the permutation test was missing from the Methods section.

Action taken: We have now created a new Figure 4—figure supplement 3 to show the distribution, and reference it from the main text. We also now include a verbal description in the Methods section.

2) The authors argue that "extra-retinal processing" is responsible for the minor shift in the center of the highest gain OKR spot. However, while the effect indeed seems likely to require extra-retinal input, the authors provide no evidence that the processing mediating this should occur outside of the retina. Given other examples of non-visual information being sent to the retina, it is at least plausible that retinal processing is involved.– Line 79: The evidence for it being "extra-retinal processing" is lacking. The source of the effect is likely extra-retinal but whether the processing happens in the retina or elsewhere is not shown.– 182: The "determinants" are indeed likely extra-retinal, but this does not mean the processing is.– Line 428: You have not demonstrated that the processing is extra-retinal, only that it is not occurring in retinal coordinates. There could be a projection into the retina, for example.

It is true that such additional retinal processing could explain the effect we observe, even though no evidence for such functional projections has so far been found. But this does indeed leave both hypotheses unfalsified.

Action taken: We have rephrased our discussion to suggest both extra-retinal processing and putative projections onto the retina as valid hypotheses. We now write:

“In addition, further processing appears to affect the preferred OKR stimulus location, as suggested by the experiments in upside-down embedded animals. This could include extra-retinal processing, or hitherto unknown projections to the retina itself.”

3) The authors should consider three coordinate systems explicitly, particularly in the lines 273-303. These would be retinal coordinates, head coordinates and environmental coordinates.

The reviewer is correct that this paragraph was not sufficiently explicit regarding the possible coordinate systems.

Action taken: We have revised the text accordingly to now include discussion of all three reference frames (see next point).

– Line 277: It seems there's also another representation which is missing from your consideration here – a body centric representation. This is distinct from retinal coordinates and distinct from environmental coordinates but is known from primate studies, for example.

Indeed, this is a crucial point we need to clarify. Our fish are immobilised, upright, and oriented towards 0 azimuth, 0 elevation. While the eyes themselves can move in their sockets, the head cannot move relative to the body. The “fish-centred coordinates” (body-centred coordinates) we discuss on line 278 are thus identical to head-centred coordinates in our experiments.

Action taken: We now present our results in the context of these four reference systems (body-, head-, retina-, and world-centred representations) in the Results section, and also added further detail in the Methods section. In the Results section, we now write:

“A priori, it is unclear whether the sampling preference originates from the peculiarities of the sensory periphery in the eye, or the behavioural relevance inferred by central brain processing. […] To start distinguishing these possible…” as well as:

“Eye-, head-, or body- centred reference frames are therefore not sufficient to fully explain the observed optokinetic stimulus location preferences. Instead, the stimulus location preference is additionally related to the environmental reference frame, suggesting that the behavioural relevance of our motion stimuli depends on their environmental elevation positions.”

In the Methods section, we added:

“Geographic coordinates can be used to identify stimulus position in at least four distinct reference frames: body-centred, head-centred, retina-centred, or world-centred coordinates. […] Therefore, our comparison of the OKR gain map to retinal photoreceptor densitiy distributions is based on the time-averaged eye position during optokinetic stimulation.”

– Line 282. Now retina-centered is missing. (the retina can move independently from the head/body).

The reviewer is correct that we didn’t discuss a possible retina-centered reference frame at this position. The eye position varies continuously during OKR, while the body – at least in our immobilised fish – remains stationary. To convincingly show OKR gain maps in a retina-centred reference frame, the stimulus position should ideally be updated in the experiment based on the current eye position, which we didn’t do. We did find, however, that the average eye position (the beating field) does not depend on stimulus position as described in the manuscript. Therefore, across different stimulus phases, the retina-centred reference frame is similar to the bodycentred reference frame.

Our discussion of fish- and world-centred coordinates on line 282 is in the context of identifying stimulus locations that evoke maximum OKR gain. We later identify the part of the retina that was stimulated by this “optimal” stimulus while the eye was at its average position. And we observe that this retinal location – as well as its surroundings that would be stimulated when the eye is outside its average position – correspond to high photoreceptor densities.

Action taken: We now discuss all possible reference systems explicitly in the Results section (also see previous point).

4) Especially as we consider this manuscript rather an advancement for the "Tools and Resources" section, please consider that the Stowers J et al. 2017 "Virtual Reality for Freely Moving Animals" was a full-surround setup – at least as defined by the standard for the other citations – used successfully on *Drosophila* and, importantly here, fish. You need to be more specific in your wording (e.g. to limit statements to systems based on LEDs capable of producing the desired spectral content) or cite that paper.

The published method suggested by the reviewer is indeed a valuable contribution and should be discussed explicitly in our manuscript. The FreemoVR platform is highly versatile, but differs from our solution in several aspects that are important to studies of zebrafish visual function (visual field coverage, light refraction, interoperability with laser-scanning microscopes). This merits a qualified discussion that was so far absent from our manuscript.

Action taken: We now included a reference to Stowers et al. 2017, and briefly compare and contrast it to our own approach. We write:

“At least one existing solution for stimulating and tracking freely moving zebrafish supports unusually large stimuli, though despite its versatility, it still only covers part of the visual field and does not address the remaining issues of total internal reflection and the interoperability with laser-scanning microscopes (43).”

5) The statement that the gain of the OKR broadly follow the density of the longer-wavelength sensitive photoreceptors in the retina is entirely ignoring the fact that the fish retina is full of Opsins in all kind of different cells types. This includes different melanopsins, but also VA-opsins (which are likely sensitive in the green), opn5s (UVA), tmt-opsins etc. Evidence from mouse and humans starts to suggest that even in these mammals- which have a highly reduced number of non-rod, non-cone opsins- melanopsin+ RGCs are involved in vision (see Annu Rev Vis Sci. 2020 Sep 15;6:453-468. doi: 10.1146/annurev-vision-030320-041239 for a recent review). This aspect needs to be addressed in their optokinetic response gain covariant analyses.

This is indeed an interesting aspect gaining increased recognition in the field. While the good match between a higher density of longer-wavelength photoreceptors and higher OKR gains could be explained without regard to other, previously overlooked opsins, we should still address their potential role in our manuscript.

Action taken: We have now expanded our discussion to acknowledge the putative role of other opsin-expressing retinal neurons. We write:

“Recent studies suggest that melanopsin-expressing ganglion cells also contribute to form vision (59, 60), and melanopsins as well as a plethora of other opsins are expressed in the zebrafish retina and brain (61, 62). It is therefore possible that photoreception via opsin-expressing neurons in the inner retina contributes to zebrafish OKR as well.”

6) Spectral information on the LEDs needs to be provided (in photons per area and time).

Yes, we agree that information on the brightness of the stimulus should be included. This will supplement the provided spectral information regarding peak wavelength and LED type in the manuscript and thereby facilitate comparison to the results of future studies.

Action taken: We have measured the luminance (10.94 ± 0.59 𝑐𝑑 𝑚^−2^) and now report it in the Methods section.